# SciBench: Evaluating College-Level Scientific Problem-Solving Abilities of Large Language Models

## Abstract

Recent advances in Large Language Models (LLMs) have demonstrated notable progress on many mathematical benchmarks. However, most of these benchmarks only contain problems grounded in junior and senior high school subjects, contain only multiple-choice questions, and are confined to a limited scope of elementary arithmetic operations. To address these issues, this paper introduces an expansive benchmark suite SciBench that aims to systematically examine the reasoning capabilities required for solving complex scientific problems. SciBench contains two carefully curated datasets: an open set featuring a range of collegiate-level scientific problems drawn from mathematics, chemistry, and physics textbooks, and a closed set comprising problems from undergraduate-level exams in computer science and mathematics. Based on the two datasets, we conduct an in-depth benchmarking study of five representative LLMs with various prompting strategies. The results reveal that current LLMs fall short of delivering satisfactory performance, with the best overall score of merely 35.80%. Furthermore, through a detailed user study, we categorize the errors made by LLMs into ten problem-solving abilities. Our analysis indicates that no single prompting strategy significantly outperforms the others and some strategies that demonstrate improvements in certain problem-solving skills could result in declines in other skills. We envision that SciBench will catalyze further developments in the reasoning abilities of LLMs, thereby ultimately contributing to scientific research and discovery.[1]

## 1 Introduction

Recent advancements in Large Language Models (LLMs) have dramatically expanded the boundaries of artificial intelligence [1–8]. They have demonstrated outstanding performance in many mathematical reasoning tasks that are typically considered challenging even for well-educated individuals [9–13]. Notably, GPT-4 achieves a remarkable score of 163 out of 170 on GRE Quantitative Exam, placing it at the 80th percentile ranking [3].

While the remarkable improvements in these benchmark performances might suggest that LLMs are capable of performing mathematical reasoning tasks, we argue that this assertion might be overly optimistic due to the inherent limitations of the current benchmarks. Firstly, many existing benchmarks such as ScienceQA [14] and GSM8K [15] only contain problems grounded in grade-level subjects, thereby lacking enough complexity. Although other benchmarks like MATH [16] introduce high-school level problems, they only involve a restricted range of operations — addition, subtraction, multiplication, and exponentiation — which do not adequately assess the depth of reasoning abilities of LLMs. Secondly, recent works including MMLU [17], AGIEval [18], and CEval [19], despite introducing challenging problems that span a wide range of disciplines, mainly focus on multiple-choice questions without providing detailed solutions. This setup could inadvertently mislead benchmark evaluation, as it allows LLMs to guess the answers from candidate choices and appear knowledgeable in comprehending the questions. Moreover, the lack of detailed solutions prevents us from understanding the limitations of LLMs and discerning why they commit certain errors. Furthermore, these benchmarks often source problems from online material, where questions are

---

[1]Code and data can be accessed at this link.

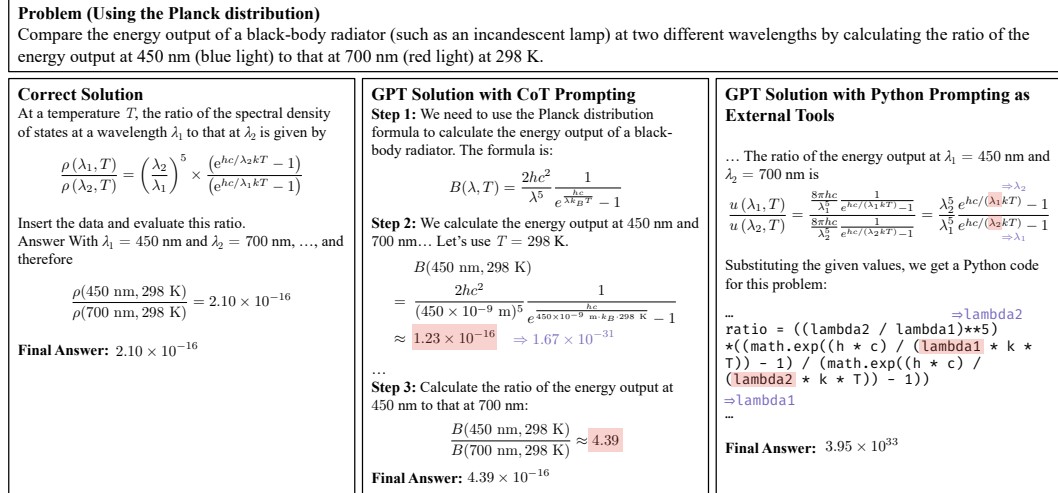

Figure 1: An example problem from *Physical Chemistry* with solutions generated under two prompting strategies. GPT-4 with Chain-of-Thought (CoT) prompting shows calculation errors, while GPT-4 that prompts Python as external tools misunderstands mathematical equations. Errors are highlighted in red and the corrections are shown in purple.

closely followed by answers. As these problems could already be a part of the training data, the models, trained in an autoregressive manner, may directly predict the answer without genuinely understanding the problem. This potential data leakage provides a shortcut for LLM evaluation, further compromising its validity.

On the other hand, many studies propose various prompting strategies aimed at enhancing the reasoning abilities for mathematical problem solving. For example, the representative strategy chain-of-thought (CoT) instructs LLMs using specific examples to generate step-by-step solutions that prompt deeper problem thinking [9, 20–22], while other strategies propose to enable LLMs to utilize external tools [23, 24] that improve the numerical computation capability. However, even these strategic approaches, each with its specific strengths, struggle to fully address complex scientific problems. Consider an example problem from college-level *Physical Chemistry* [25] that requires the use of the Planck distribution to derive certain quantities. As shown in Figure 1, LLMs with CoT prompts accurately generate the correct formula, but fail in the final numerical calculation. Further, when explicitly instructed to generate a Python program to solve this problem alongside the reasoning process of CoT, the LLM derives an incorrect equation, misplacing $\lambda_1$ in the numerator rather than the denominator. This error illustrates that LLMs struggle to comprehend mathematical relationships when employing external tools. This example underscores the need for a fine-grained analysis of the essential skill set required for complex scientific problem solving.

To mitigate the aforementioned deficiencies in existing LLM evaluation, this paper introduces a novel college-level **Sci**entific problem solving **Bench**mark, referred to as SCIBENCH. Our SCIBENCH contains two datasets of college-level scientific problems. The open dataset includes 695 problems collected from widely-used textbooks in college-level Chemistry, Physics, and Math courses. To simulate real-world evaluation, we also include a closed dataset that encompasses seven sets of midterm and final examination questions from three college courses in Computer Science and Mathematics. Distinct from existing benchmarks, all of the problems in SCIBENCH are open-ended, free-response questions. They require multiple steps of reasoning and the computation therein involve complex arithmetic operations such as differentiation and integration. To ensure the integrity of our evaluation, these datasets have been manually extracted from PDF documents and formatted into LaTeX documents, thereby minimizing the possibility of their leakage in LLM training data. Importantly, SCIBENCH also includes detailed solution steps, facilitating detailed error analysis.

Our evaluation includes five representative LLMs: two open-source models LLaMA-2-7B and LLaMA-2-70B, and three close-source models Claude2, GPT-3.5, and GPT-4, with various prompting strategies, including CoT, zero-shot learning, and few-shot learning. In addition, we also prompt LLMs to utilize external tools such as Python and Wolfram languages. The experimental results indicate that the complexity and difficulty of our dataset are sufficient to differentiate the performance

Table 1: Comparison of SCIBENCH with other benchmarks. "Level" represents the grade level of problems. "Computation" represents the level of computational type that problems use. "Solution" represents whether datasets contain detailed solutions. "Type" represents the type of most problems provided in the dataset: "MT" denotes multiple-choice questions and "Free" denotes free-response questions. "Human" indicates whether the analysis process employs a human annotation process. "Auto" represents whether the analysis process uses an automatic annotation process.

| Benchmark | Dataset | | | | Evaluation | | | | Analysis | |
| --- | --- | --- | --- | --- | --- | --- | --- | --- | --- | --- |
| | Level | Computation | Solution | Type | Zero-Shot | Few-Shot | CoT | Tool | Human | Auto |
| ScienceQA [14] | Grade 1–12 | Algebra | Yes | MT | Yes | Yes | Yes | No | No | No |
| IconQA [26] | Grade 1-12 | Algebra | No | MT | No | Yes | No | No | No | No |
| TabMWP [27] | Grade 1–12 | Algebra | Yes | Free | No | Yes | No | No | No | No |
| GSM8K [15] | Grade 1–12 | Algebra | Yes | Free | No | Yes | No | No | No | No |
| MATH [16] | High School | Exponentiation | Yes | Free | No | Yes | No | No | No | No |
| LILA [28] | High School | Exponentiation | Yes | Free | Yes | Yes | No | No | No | No |
| SciEval [29] | High School | Exponentiation | No | MT | Yes | Yes | Yes | No | No | No |
| MMLU [17] | High School + College | Exponentiation | No | MT | No | Yes | No | No | No | No |
| CEval [19] | High School + College | Differentiation | No | MT | No | Yes | Yes | No | No | No |
| AGIEval [18] | High School + College | Exponentiation | No | MT | Yes | Yes | Yes | No | Yes | No |
| TheroemQA [30] | College | Differentiation | No | Free | No | Yes | Yes | Yes | No | No |
| SCIBENCH | College | Differentiation | Yes | Free | Yes | Yes | Yes | Yes | Yes | Yes |

levels of different LLMs. With the strongest configuration, which combines both CoT prompting and external tools, GPT-4 achieves an average score of 35.80% on the open dataset and 51.57% on the closed exam dataset. These results suggest a considerable potential for improvement in future LLMs.

In order to gain a comprehensive understanding of the limitations of LLMs in scientific problem solving, we propose a novel self-refinement method to uncover the deficient skills in the solutions made by LLMs. Firstly, we compare the correct solutions with the solutions generated by LLMs and, with the assistance of human annotators, summarize ten essential skills requisite for successful scientific problem-solving. These skills include proficiency in domain knowledge, mathematical reasoning, numerical calculation abilities, and comprehension of common sense concepts. Subsequently, we employ an LLM-empowered self-critic approach to automatically classify the lacking skills in the solutions made by the benchmarked LLMs under each experiment configuration. Our analysis finds that (1) although CoT significantly improves the calculation ability, it is less effective in other aspects; (2) prompts with the use of external tools could potentially compromise the other fundamental skills; (3) few-shot learning does not universally improve scientific problem-solving skills.

## 2 RELATED WORK

Recently, many benchmarks focus on assessing problem-solving skills of LLMs, particularly in scientific and mathematical domains [17, 18, 27, 28, 30–33]. GSM8K [15] is a widely used math dataset containing 8.5K grade school math word problems. ScienceQA [14] is a multimodal question-answering dataset with accompanying lecture and explanation annotations. MATH [16] presents a challenging collection of 12.5K math problems gathered from math competitions. LILA [28] extends 20 datasets by including task instructions and Python program solutions. However, the majority of those benchmarks concentrates on the grade or high school level tasks involving basic arithmetic operations such as addition, multiplication, and exponentiation, rather than more sophisticated operations like differentiation. TheroemQA [30] is a theorem-oriented dataset comprising 800 high-quality questions that aim to evaluate the ability of LLMs to apply theorems to solve problems. However, it lacks an in-depth qualitative analysis of their benchmark. Galactica [34] provides a set of scientific tasks, including LaTeX equation conversions, domain knowledge probes, citation prediction and chemical QA. BIG-Bench [35] is a large-scale general-purpose test suite comprising 204 multiple-choice or exact-match tasks, while BIG-Bench [36] Hard poses particularly challenging chain-of-thought prompts. C-EVAL [19] focuses on evaluating LLMs in Chinese, offering questions from humanities to science and engineering. SciEval [29] includes a mix of objective and subjective questions across multiple scientific fields to assess understanding, application, and research capabilities. AGIEval [18] evaluates the performance of LLMs in human-centric standardized exams, such as college entrance exams and lawyer qualification tests. It also provides human annotated qualitative analysis to analyze the capabilities of the model. However, relying on human labor for direct solution analysis can be costly. Our evaluation protocol, based on predefined fundamental problem solving skills, enables automated classification of deficient skills for each incorrectly answered question. This

Table 2: Summary of the open textbook dataset. We report the number of problems and the ratio of problems with detailed solutions in the fourth and fifth columns respectively.

| Subject | Title | Acronym | # Problems | % Solutions |
|---|---|---|---|---|
| Physics | *Fundamentals of Physics* [37] | `fund` | 83 | 12.0% |
| | *Statistical Thermodynamics* [38] | `thermo` | 84 | 20.2% |
| | *Classical Dynamics of Particles and Systems* [39] | `class` | 54 | 13.0% |
| Chemistry | *Quantum Chemistry* [40] | `quan` | 42 | 19.0% |
| | *Quantum Chemistry* [41] | `chemmc` | 48 | 18.8% |
| | *Physical Chemistry* [42] | `atkins` | 123 | 13.0% |
| | *Physical Chemistry, Quanta, Matter, and Change* [25] | `matter` | 59 | 16.9% |
| Math | *Calculus: Early Transcendentals* [43] | `calc` | 52 | 19.2% |
| | *Probability and Statistical Inference* [44] | `stat` | 95 | 21.1% |
| | *Elementary Differential Equations and Boundary Value Problems* [45] | `diff` | 55 | 9.1% |

approach enables an affordable, large-scale of qualitative analysis over model solutions. We include the comparison between different benchmarks in Table 1.

## 3 THE SCIBENCH DATASET

To evaluate the capabilities and analyze the limitations of Large Language Models (LLMs) to solve scientific computing problems, we collect a new dataset consisting of college-level textbooks and course exams in a variety of domains. This section details the dataset construction process.

**Data selection.** Our dataset aims to improve the previous benchmarks by including more challenging problems, which require more reasoning steps, and more advanced types of computations. Specifically, the selected dataset should fulfill the following requirements:

- **Inclusion of college-level problems.** The chosen problems demand a solid understanding of domain-specific knowledge, proficiency in reasoning capability, adept calculation skills, and the ability to comprehend complex concepts.

- **Inclusion of detailed solutions.** To facilitate a thorough analysis of the limitations of LLMs, detailed solutions should be provided as well, which could facilitate a finer-grained examination of the capacity of LLMs to handle complex problem-solving tasks.

- **Inaccessibility in text formats.** To ensure an unbiased evaluation, questions should not be readily accessible online and cannot be easily extracted or transformed into text. This aims to mitigate any potential information leakage from the exposure of LLMs to pre-existing online question banks, such as those found in standardized tests like the SAT exams.

- **Enabling of assessing advanced problem solving ability.** The problems to benchmark should not be confined to basic arithmetic operations like addition and multiplication. Rather, they should enable evaluating the capability of LLMs in performing advanced computations such as integration and differentiation, particularly when dealing with exceptionally small or large floating numbers.

Accordingly, we select ten textbooks that have been extensively used in college courses as the open textbook dataset from three scientific fields Physics, Chemistry, and Math. We report the number of problems and the ratio of problems with detailed solutions of each title in Table 2. For brevity, we will be using their acronyms when referring to specific textbooks throughout the paper. Furthermore, in order to simulate real-world evaluation, we collect a closed set of exam questions from college courses from Computer Science and Math departments, including *Data Mining*, *Machine Learning*, and *Differential Equations*. The statistics of the problems in each exam is detailed in Table 3. We refer readers of interest to Appendix A for details on these textbooks and exams.

To reduce the likelihood of correct answers being merely guessed from candidates, we choose to mainly include questions with more challenging, free-response answers, rather than multiple-choice questions in previous works [14, 30, 46]. In order to facilitate standardized and automated evaluation, we focus on answers that only contain single numerical numbers to avoid ambiguity for the textbook dataset. Further, we convert the answer to floating-point numbers rounded to three decimal places. For example, the answer $\frac{\sqrt{2}}{\pi}$ will be converted to the decimal representation of $0.450$. We also treat scientific notation as a unit to avoid overflow issues. For example, if the answer is $2.2 \times 10^{-31}$ m, we take $2.2$ as the final answer and $10^{-31}$ m as the unit.

Table 3: Statistics of the close exam dataset. We report the number of problem instances in each exam and the ratio of problems in the exam that include detailed solutions. We further report the ratio of problems in different formats, including free-response, multiple-choice, and true-false. For reference, the number in parentheses denotes the grading points assigned to the problems.

|  | Data Mining | | Machine Learning | | Differential Equations | | |
|---|---|---|---|---|---|---|---|
|  | Midterm | Final | Midterm | Final | Exam 1 | Exam 2 | Final |
| # Problems | 25 (90) | 24 (75) | 12 (56) | 16 (75) | 8 (100) | 8 (100) | 11 (95) |
| % Solutions | 56.0% (58) | 16.7% (19) | 100.0% (56) | 31.2% (26) | 100.0% (100) | 100.0% (100) | 90.9% (90) |
| % Free-response | 40.0% (46) | 33.3% (29) | 66.7% (38) | 81.3% (62) | 100.0% (100) | 100.0% (100) | 90.9% (90) |
| % Multiple-choice | 28.0% (28) | 29.2% (28) | 33.3% (18) | 18.7% (13) | 0.0% (0) | 0.0% (0) | 9.1% (5) |
| % True-false | 32.0% (16) | 37.5% (18) | 0.0% (0) | 0.0% (0) | 0.0% (0) | 0.0% (0) | 0.0% (0) |

**Data preprocessing.** We collect each problem from the original textbooks in PDF documents and manually process them into LaTeX documents using an OCR tool Mathpix. The data is manually collected by human annotators using a web-based annotation tool [46], whose user interface is shown in Appendix B. All problems are carefully verified by human annotators to ensure that LaTeX documents can be compiled without any syntax errors. For reference, we also provide the original numbers in textbooks. For every problem, we provide the answer in two forms: the numerical value and the corresponding LaTeX expression with mathematical notations retained (e.g., $0.450$ and $\frac{\sqrt{2}}{\pi}$); the unit of each answer is saved as a separate attribute. The detailed step-by-step solutions are also provided in LaTeX. For problems having multiple answers, we either keep only the first subproblem and discard the remaining subproblems or convert each subproblem into a separate problem.

## 4 EXPERIMENTS

### 4.1 EXPERIMENT SETUP

We evaluate three close-source LLMs: Claude2 (`claude2`) [47], GPT-3.5 (`gpt-3.5-turbo`) [2], GPT-4 (`gpt-4`) [3], along with two open LLMs: LLaMA-2-7B (`llama-2-7b-chat`) and LLaMA-2-70B (`llama-2-70b-chat`) [48] on two benchmark datasets. We consider two prompting strategies, including the Chain-of-Thought (CoT) prompting and prompting to use external tools, under both zero-shot and few-shot learning paradigms.

- **Zero-shot and few-shot learning.** In the zero-shot learning setting, models are not provided with any prior examples, which evaluates their inherent problem-solving capabilities with background knowledge and reasoning abilities. In the few-shot setting, a few of examples are given to the models before the test example. This aims to assess their capability to learn new information from the demonstrations and incorporate it into their problem-solving processes.

- **Prompting-based approaches.** In the zero-shot setting, we evaluate both with and without the system prompt, which describes the types and categories of questions, along with instructions; all other settings incorporate the system prompt. Additionally, we utilize CoT as our prompting strategy in the zero-shot setting. Besides, we further explore an answer-only strategy in the few-shot setting, where the prompt solely provides questions and answers without any intermediate solutions.

- **Tool-augmented approaches.** Given that LLMs are limited to acquiring exact knowledge and performing precise calculations, some recent approaches, such as Toolformer [23] and Chameleon [24], explored the use of external tools to enhance the capabilities of solving complex reasoning tasks. In line with this approach and acknowledging the limitations of LLMs in performing precise calculations, we also include a setting that prompts the model to convert its solution steps in natural language into either Wolfram Language[2] or Python code, aiming to achieve more accurate results for certain computation steps. This prompt is only tested in the few-shot learning setting. We manually construct Python and Wolfram Language code that produces the correct answer.

In summary, we consider seven combinations of prompting strategies and learning paradigms: zero-shot learning without the system prompt (*Zero−S*), zero-shot learning with the system prompt (*Zero*), few-shot learning (*Few*), CoT prompting under zero-shot (*Zero+CoT*) and few-shot learning (*Few+CoT*) scenarios, few-shot learning that prompts to use Python (*Few+Py*), and Wolfram

---

[2]https://www.wolfram.com/language/

Table 4: Experimental results in terms of accuracy (%) on the textbook dataset. The best performing score is highlighted in **bold** and second-best is underlined.

| Model | Setting | Chemistry | | | | Physics | | | Math | | | Avg. |
|-------|---------|--------|--------|------|--------|------|-------|--------|------|------|------|------|
| | | atkins | chemmc | quan | matter | fund | class | thermo | diff | stat | calc | |
| LLaMA-2-7B | Zero−S | 0.00 | 0.00 | 0.00 | 0.00 | 1.37 | 0.00 | 0.00 | 2.00 | 2.67 | 4.76 | 0.60 |
| | Zero | 0.00 | 0.00 | 0.00 | 0.00 | 1.37 | 0.00 | 0.00 | 2.00 | 5.33 | 0.00 | 0.60 |
| | Zero+CoT | 0.00 | 2.56 | 0.00 | 0.00 | 0.00 | 0.00 | 0.00 | 0.00 | 4.00 | 0.00 | 0.40 |
| | Few | 3.74 | 5.13 | 5.99 | 2.04 | 4.11 | 0.00 | 1.49 | 6.00 | 8.00 | 0.00 | 2.20 |
| | Few+CoT | 1.87 | 5.13 | 2.94 | 0.00 | 5.48 | 0.00 | 0.00 | 0.00 | 12.00 | 7.14 | 2.10 |
| | Few+Py | 0.93 | 2.56 | 0.00 | 0.00 | 0.00 | 0.00 | 0.00 | 0.00 | 6.67 | 0.00 | 0.70 |
| | Few+Wol | 0.00 | 0.00 | 0.00 | 0.00 | 0.00 | 0.00 | 0.00 | 0.00 | 0.00 | 0.00 | 0.00 |
| LLaMA-2-70B | Zero−S | 1.87 | 2.56 | 0.00 | 0.00 | 0.00 | 0.00 | 0.00 | 0.00 | 2.67 | 0.00 | 0.50 |
| | Zero | 1.87 | 2.56 | 0.00 | 0.00 | 1.40 | 0.00 | 0.00 | 0.00 | 10.70 | 4.76 | 1.41 |
| | Zero+CoT | 0.93 | 2.56 | 0.00 | 0.00 | 0.00 | 0.00 | 1.49 | 0.00 | 10.70 | 0.00 | 1.10 |
| | Few | 9.30 | 12.83 | 14.71 | 2.04 | 15.07 | 6.38 | 2.94 | 8.00 | 21.33 | 9.52 | 6.09 |
| | Few+CoT | 13.10 | 12.83 | 14.71 | 4.08 | 12.33 | 0.00 | 0.00 | 0.00 | 13.30 | 9.52 | 4.90 |
| | Few+Py | 0.93 | 7.69 | 2.94 | 0.00 | 9.59 | 0.00 | 1.49 | 0.00 | 17.30 | 9.52 | 2.99 |
| | Few+Wol | 1.87 | 0.00 | 0.00 | 0.00 | 1.39 | 0.00 | 0.00 | 2.00 | 5.33 | 11.90 | 1.30 |
| Claude2 | Zero−S | 16.82 | 17.95 | 8.82 | 8.16 | 6.85 | 12.77 | 7.46 | 4.00 | 37.33 | 9.52 | 8.20 |
| | Zero | 15.00 | 12.83 | 14.71 | 10.20 | 12.33 | 6.40 | 9.00 | 4.00 | 38.70 | 16.70 | 8.71 |
| | Zero+CoT | 20.56 | 15.38 | 8.82 | 4.08 | 8.23 | 4.26 | 5.97 | 6.00 | 36.00 | 14.29 | 8.10 |
| | Few | 15.87 | 20.51 | 8.82 | 8.16 | 6.85 | 10.64 | 8.51 | 4.00 | 32.00 | 11.90 | 6.09 |
| | Few+CoT | 15.89 | 25.64 | 14.65 | 6.12 | 9.59 | 6.38 | 10.45 | 8.00 | 33.33 | 19.05 | 8.90 |
| | Few+Py | 6.54 | 12.82 | 14.71 | 4.08 | 17.81 | 8.51 | 5.97 | 20.00 | 40.00 | 16.67 | 8.70 |
| | Few+Wol | 9.35 | 0.00 | 2.94 | 0.00 | 1.39 | 0.00 | 0.00 | 2.00 | 5.33 | 11.90 | 2.20 |
| GPT-3.5 | Zero−S | 8.41 | 28.21 | 5.88 | 4.08 | 12.33 | 2.13 | 5.97 | 4.00 | 21.33 | 13.95 | 10.62 |
| | Zero | 4.67 | 20.51 | 8.82 | 2.04 | 10.96 | 2.13 | 2.94 | 6.00 | 28.00 | 9.30 | 9.59 |
| | Zero+CoT | 6.54 | 23.08 | 2.94 | 10.20 | 12.33 | 2.12 | 5.97 | 12.00 | 33.33 | 9.30 | 12.17 |
| | Few | 5.61 | 15.38 | 11.76 | 4.08 | 8.22 | 0.00 | 1.49 | 10.00 | 26.67 | 13.95 | 9.60 |
| | Few+CoT | 8.41 | 20.51 | 8.82 | 6.12 | 10.96 | 2.12 | 1.49 | 10.00 | 38.67 | 6.98 | 11.99 |
| | Few+Py | 13.08 | 33.33 | 8.82 | 16.33 | 26.01 | 4.26 | 7.46 | 16.00 | 44.00 | 26.19 | 19.91 |
| | Few+Wol | 3.74 | 7.69 | 2.94 | 18.37 | 17.81 | 6.38 | 2.99 | 12.00 | 5.33 | 2.38 | 7.87 |
| GPT-4 | Zero−S | 14.95 | 25.64 | 8.82 | 18.37 | 21.92 | 12.77 | 7.46 | 8.00 | 28.00 | 19.05 | 16.81 |
| | Zero | 27.10 | 23.08 | 14.71 | 22.45 | 15.07 | 8.51 | 11.94 | 18.00 | 56.00 | **42.86** | 25.09 |
| | Zero+CoT | **28.04** | 43.59 | 14.71 | 20.41 | 21.92 | **19.15** | 17.91 | 22.00 | 50.67 | **42.86** | 28.52 |
| | Few | 15.87 | 30.77 | 17.65 | 12.24 | 26.03 | 12.77 | 5.97 | 8.00 | 49.33 | 33.33 | 21.46 |
| | Few+CoT | 21.05 | **46.15** | 17.65 | 26.53 | 27.40 | 14.00 | 13.43 | 18.00 | 61.33 | 35.71 | 28.35 |
| | Few+Py | 21.05 | 41.03 | **38.24** | **28.57** | **38.36** | 17.02 | **29.85** | **34.00** | **69.33** | **42.86** | **35.80** |
| | Few+Wol | 3.74 | 0.00 | 17.65 | 26.53 | 27.30 | 17.02 | 17.91 | 32.00 | 7.69 | 14.29 | 15.56 |

Language (*Few+Wol*) as external tools. Regarding the exam dataset, to replicate a real-world exam environment, we only consider two specific settings: zero-shot learning (*Zero*) and zero-shot learning supplemented with CoT prompting (*Zero+CoT*).

**Implementation details.** We set temperature to zero for all models to reduce the randomness of the predictions. Few-shot examples, including solutions, are randomly selected from problems within each textbook. When external tools are used, we add a code snippet that translates the solution into specific programming languages in all few-shot examples. The code snippets are verified by human annotators that will produce the correct output. In terms of evaluation metrics, we compare the model outputs with the correct answers, allowing a relative tolerance of 0.05. In particular to the exam dataset, the model solutions are graded using the rubrics provided by the instructors. Readers may refer to Appendix C for all prompts and the implementation details for utilizing external tools.

## 4.2 RESULTS AND ANALYSIS

We report the model performance in terms of accuracy score for each textbook and an average score over all problems. The results of all LLMs in various settings on the textbook and the exam dataset are summarized in Tables 4 and 5 respectively. We have the following observations.

- **Observation 1. SCIBENCH is complex enough to differentiate among LLMs.** Our findings show that open-source models LLaMA-2-7B and LLaMA-2-70B do not yet rival closed-source counterparts on both textbook and exam datasets, where the best performance is obtained with GPT-4 with Python as the external tool in the few-shot learning setting. Within both the LLaMA and GPT series, we also observe a clear correlation between increased model capacity (i.e., larger

Table 5: Experimental results in terms of total scores under zero-shot learning on the exam dataset. The best performing score is highlighted in **bold**.

| Model | Setting | Data Mining | | Machine Learning | | Differential Equations | | |
|---|---|---|---|---|---|---|---|---|
| | | Midterm | Final | Midterm | Final | Exam 1 | Exam 2 | Final |
| LLaMA-2-7B | Zero | 24 / 90 | 14 / 75 | 6 / 56 | 6/ 75 | 5 / 100 | 0 / 100 | 0 / 95 |
| | Zero+CoT | 18 / 90 | 14 / 75 | 2 / 56 | 10 / 75 | 10 / 100 | 0 / 100 | 10 / 95 |
| LLaMA-2-70B | Zero | 23 / 90 | 18 / 75 | 18 / 56 | 12 / 75 | 20 / 100 | 5 / 100 | 0 / 95 |
| | Zero+CoT | 31 / 90 | 18 / 75 | 10 / 56 | 11/ 75 | 35 / 100 | 10 / 100 | 0 / 95 |
| Claude2 | Zero | 37 / 90 | 26 / 75 | 28 / 56 | 35 / 75 | 35 / 100 | 30 / 100 | 20 / 95 |
| | Zero+CoT | 33 / 90 | 38 / 75 | 22 / 56 | **41** / 75 | 25 / 100 | 15 / 100 | 20 / 95 |
| GPT-3.5 | Zero | 44 / 90 | 39 / 75 | 16 / 56 | 32 / 75 | 0 / 100 | 45 / 100 | 15 / 95 |
| | Zero+CoT | 38 / 90 | 33 / 75 | 32 / 56 | 37 / 75 | 28 / 100 | 30 / 100 | 10 / 95 |
| GPT-4 | Zero | 56 / 90 | **44** / 75 | 30 / 56 | 37 / 75 | 25 / 100 | **80** / 100 | **25** / 95 |
| | Zero+CoT | **58** / 90 | 32 / 75 | **40** / 56 | 35 / 75 | **50** / 100 | 70 / 100 | 15 / 95 |

parameter sizes) and improved performance. This observation demonstrates that the complexity of SCIBENCH is able to differentiate the capacities of different LLMs.

- **Observation 2. The zero-shot learning setting exhibits comparable performance to the few-shot learning setting.** For example, with CoT prompting, Claude2 achieves average scores of 8.10% and 8.90%, and GPT-4 achieves 28.52% and 28.35% in zero- and few-shot settings respectively. Moreover, in many textbooks such as *Quantum Chemistry* (`quan` and `chemmc`), which focus on a specialized subdomain within each field, few-shot learning outperforms zero-shot learning, with improvements of 2.94% and 2.56% in GPT-4 and 14.7% and 10.20% in LLaMA-2-70B under the CoT setting, for instance. This could be attributed to the selected prompt examples being representative and informative to the domain.

- **Observation 3. Utilizing advanced prompting strategies like CoT brings advantages over vanilla LLMs.** For the textbook dataset, the CoT prompting yields average improvements of 2.58% and 2.39% under zero-shot and few-shot learning for GPT-3.5, and 3.43% and 6.89% for GPT-4, respectively. This improvement suggests that encouraging LLMs to generate detailed solution steps helps obtain correct final answers, though its effectiveness varies across different models and settings. However, this trend is less obvious in LLaMA models with 6.09% and 4.90% in LLaMA-2-70B under the few-shot setting, possibly due to their inherent inadequacy.

- **Observation 4. Prompts that utilize Python yield improvements in certain models while those using Wolfram diminish performance.** Under few-shot learning scenarios, utilizing Python as an external tool results in an improvement of 7.92% compared to the CoT prompting for GPT-3.5, and an improvement of 7.45% for GPT-4. However, in Claude2, this trend is less evident with average scores of 8.90% and 8.70% with and without utilizing Python. Similarly, LLaMA models exhibit a decrease in performance from 4.90% to 2.99% in LLaMA-2-70B.

  Utilizing Wolfram Language does not help few-shot learning and even results in a deteriorated performance, with a decrease of 6.70% compared to the CoT prompting for Claude2, and a decrease of 12.79% for GPT-4. We note that converting the solution steps to Wolfram Language often introduces syntax issues and thus fails to produce satisfactory results, particularly in textbooks like *Quantum Chemistry* (`chemmc`), which involve numerous variables.

## 5 ERROR ANALYSIS OF VARIOUS PROMPTING STRATEGIES

Considering the substantial advancements of current LLMs, an in-depth analysis of the particular skills that are either enhanced or limited under certain settings becomes imperative. Previous works have relied on human labor to annotate error reasons into different categories, which is both expensive and time-consuming [18]. In this section, we present an evaluation protocol that automates the classification of error reasons into deficient skills. This time-efficient approach enables large-scale analyses in future research.

In order to quantify the impact of each setting on scientific problem-solving, we first define an essential skill set that is required by solving scientific problems. Then, an LLM verifier is employed to automatically classify each incorrectly solved problem based on the absence of a specific skill

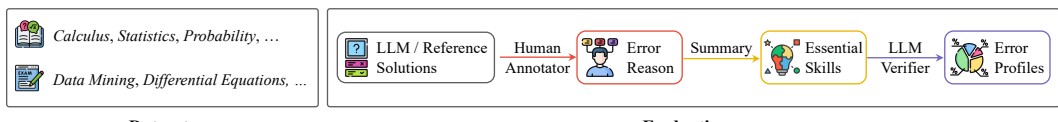

**Datasets**                                              **Evaluation**

Figure 2: Pipeline of the evaluation protocol. The evaluation protocol involves analyzing both LLMs and reference (correct) solutions with the assistance of human annotators to identify error reasons. These reasons are then summarized into ten essential scientific problem-solving skills in which LLM may face challenges. Subsequently, a LLM verifier is employed to automatically attribute each incorrectly answered problem to a lack of a specific skill. The resulting error profiles enable the interpretation of the improved skills by certain prompting strategies and the direct comparison of various strategies.

from the essential skill set. This approach generates error profiles, showcasing a direct comparison of different strategies. This evaluation protocol is summarized in Figure 2.

Firstly, we analyze the incorrect solutions made by GPT-3.5 for problems that provide detailed solutions. We hire two college students, who are highly familiar with the problems in our datasets, to annotate the source of the error for each problem, indicating the specific line where the model makes a mistake and why. From 112 such error annotations and with the assistance of GPT-4, we distill these errors into ten essential skills that GPT-3.5 might lack:

- **Logical decomposition and analysis skills.** This ability involves decomposing the problem into smaller, manageable parts, and understanding the relationships between these parts.

- **Identification of assumptions.** This skill involves the ability to recognize relevant and necessary assumptions in the problem.

- **Spatial perception.** This is important for understanding problems in areas such as Physics and Chemistry, where models need to visualize molecules, forces, fields, etc.

- **Causal reasoning.** This is the ability to understand cause and effect relationships.

- **Problem deduction skills.** This pertains to the ability to infer and deduce potential solutions or underlying principles from the given information in a problem.

- **Abstract reasoning.** This skill involves the ability to understand complex concepts that cannot be perceived physically, and to recognize patterns or relationships beyond concrete examples.

- **Scientific literacy.** This skill involves a comprehensive understanding of key scientific principles, terminology, and methodologies across a range of disciplines.

- **Code conversion skills.** This involves the ability to accurately translate solution steps into different programming languages, like Python or Wolfram Language.

- **Logical reasoning.** This is the ability to make a reasoned argument and to identify fallacies or inconsistencies in an argument or set of data.

- **Calculation skills.** This involves the ability to accurately carry out mathematical operations and computations.

After identifying this essential skill set, we assess the performance of the LLMs under different settings to discern the specific problem-solving skills they lack. Given the high cost of human annotations required to attribute the cause of incorrect solutions to specific skill deficiencies, we propose a novel self-critique protocol: we design a specific prompt that outlines these abilities, and employ another LLM to serve as a classifier and determine whether a specific error results from the lack of a particular problem-solving skill. Finally, we ask human annotators to scrutinize the classification results, which results in approximately 20% of incorrectly classified skills being discarded. To be specific, we utilize a GPT-3.5 model as the verifier to determine the reason behind each error and pinpoint the missing skill. The details regarding the specific prompts used are provided in Appendix C.1. This verification process is conducted for six settings, with results represented in bar charts (Figure 3). Additional examples of the evaluation protocol are elaborated in Appendix D.

Overall, our findings suggest that **there is a lack of a universally effective setting: each configuration only enhances some specific abilities and occasionally even hurts other skills that the original GPT models possess.** First, CoT prompting significantly improves calculation skills in

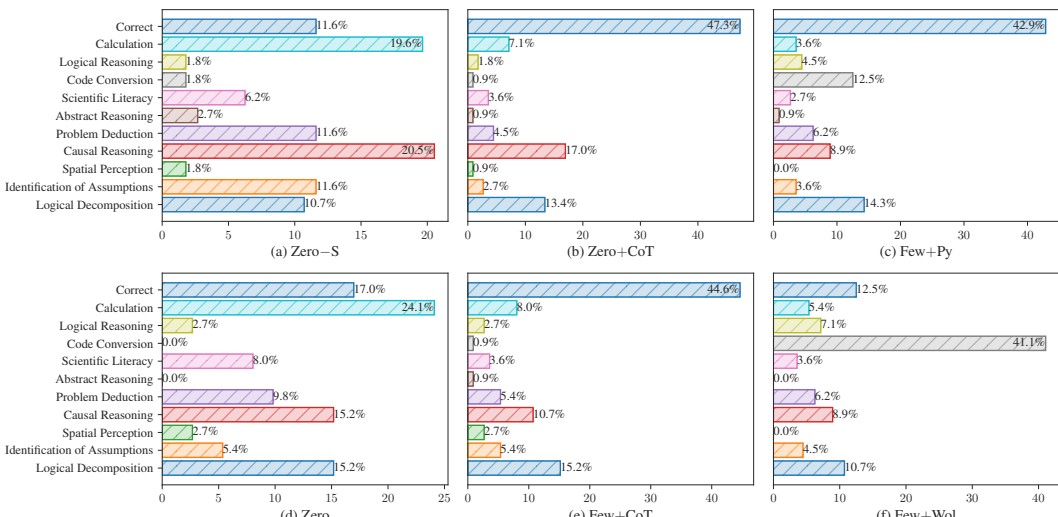

Figure 3: Error profiles of GPT-3.5 on the text dataset under six settings, which reveal the distribution of their deficiencies in ten essential problem-solving abilities.

both zero- and few-shot scenarios, with 7.1% and 8.0% error rates caused by calculation ability respectively, considerably lower than the 24.1% error rate of the vanilla zero-shot baseline. However, CoT shows limitations in improving other skills, with 15.2% error rates in both casual ability and logical decomposition ability in the zero-shot CoT setting, respectively, compared to 17.0% and 13.4% in the zero-shot setting. This contradicts previous claims about universal skill enhancement through zero-shot CoT and carefully-designed few-shot CoT prompts [9]. In Appendix, we show an example in Figure S3, where the zero-shot learning setting without CoT has generated the correct formula but fails in the calculation steps. In this case, CoT prompting is even unable to use the correct formula as it misinterprets the specific conditions (non-necessity) in the problem. Second, while the use of external tools significantly reduces calculation errors, they can weaken other skills, particularly the code conversion skills, i.e., generating the correct programs for the solution. This issue becomes particularly prominent when using the Wolfram Language, with 41.1% error rate in code conversion skill comparing 0.9% in the few-shot CoT setting. Despite providing grammar specifications in system prompts and a few examples as demonstrations, most attempts of code conversion result in syntax errors. In Wolfram Language, the error mainly comes from the violation of variable rules (for instance, Wolfram Language reserves certain letters such as $E$ as protected symbols and disallows underscores in variable names) or incorrect usage of certain functions.

Additionally, **few-shot learning does not universally improve scientific problem-solving skills**, as indicated in the comparison between zero-shot and few-shot CoT settings. The improvement in one skill is offset by the shortcomings in others: although the few-shot CoT setting results in a reduction of 6.3% in errors related to causal reasoning, it also leads to an increase in errors associated with other skills, such as logical decomposition and calculation.

Moreover, **the skill of identifying assumptions appears to be most lacking in the zero-shot setting without a system prompt**. In this scenario, the LLM does not have any predefined direction to follow. However, when a system prompt with instructions about which scientific domain the model is tackling, this issue can be significantly mitigated, decreasing this error from 11.6% to 5.4%.

## 6 CONCLUSION

In conclusion, this paper presents SCIBENCH, a college-level dataset that includes scientific problems from Mathematics, Physics, and Chemistry, as well as exam questions in Computer Science and Mathematics. We also conduct extensive experiments on five representative models, LLaMA-2-7B, LLaMA-2-70B, Claude2, GPT-3.5, and GPT4. The evaluation protocol we employ serves as a framework for evaluating advanced problem-solving skills of LLMs in scientific domains. The findings of this study highlight that while large language models (LLMs) exhibit impressive performance on introductory mathematical benchmarks, their mastery of problem solving ability

remains weak. These findings underscore the limitations of current LLMs in achieving satisfactory performance, even with the assistance of various tools. We envision that the SCIBENCH benchmark dataset and evaluation protocol presented in this paper could lay a foundation for future research and enable advancements in understanding and enhancing problem-solving capabilities of LLMs.

## REPRODUCIBILITY STATEMENT

To foster reproducible research, we include all dataset processing and experiment details of SCIBENCH. We detail data processing in Section 3 and provide the UI design of data collection in Appendix B. We include all experiment details with LLM prompts in Appendix C. Finally, we make our dataset and code publicly available at this anonymous repository.

## ETHICAL STATEMENT

The questions of SCIBENCH are sourced from science textbooks and exams. We conduct a manual examination of our dataset to ensure the absence of potential sensitive background or ethical concerns. The inclusion of exam questions has been authorized by the instructors of the respective courses. To the best of our knowledge, there are no ethical concerns or sensitive information present in the dataset.

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

# Supplementary Material for SCIBENCH

## A   SCIBENCH: TEXTBOOK SOURCES

### A.1   TEXTBOOK

- PHYSICAL CHEMISTRY, ATKINS ET AL. [42] (atkins) provides an exploration of equilibrium, structure, and reactions, integrating contemporary techniques like nanoscience, spectroscopy, and computational chemistry.

- QUANTUM CHEMISTRY, MCQUARRIE [41] (chemmc) meticulously covers Quantum Mechanics, from foundational principles like blackbody radiation and Heisenberg's Uncertainty Principle to complex topics such as Schrödinger's equation, quantum mechanical operators, and the application of quantum mechanics in chemical bonding.

- QUANTUM CHEMISTRY, LEVINE ET AL. [40] (quan) explores quantum chemistry, providing a detailed understanding of the Schrödinger equation, particle behavior in various scenarios, quantum mechanics operators, and other foundational quantum principles. It delves into specific applications like the electronic structure of diatomic and polyatomic molecules, variation methods, perturbation theory, electron spin and its implications in quantum mechanics, as well as various computational methods for molecular quantum mechanics.

- PHYSICAL CHEMISTRY, QUANTA, MATTER, AND CHANGE, ATKINS ET AL. [25] (matter) combines physics and mathematics, beginning with basics like differentiation and integration, advancing through quantum mechanics and atomic structure, then exploring thermodynamics, molecular motion, and chemical kinetics. Each section is supplemented with mathematical concepts such as differential equations, vectors, and probability theory.

- CLASSICAL DYNAMICS OF PARTICAL AND SYSTEMS, THORNTON AND MARION [39] (class) initiates with an exploration of fundamental mathematical concepts, discussing scalars, vectors, matrix operations, coordinate transformations, differentiation, and integration of vectors, using these constructs to illustrate concepts like velocity, acceleration, and angular velocity. It then transitions into the realm of Newtonian mechanics, detailing Newton's laws, frames of reference, and the equation of motion for a single particle.

- THERMODYNAMICS, STATISTICAL THERMODYNAMICS, AND KINETICS, [38] (thermo) navigates through thermodynamics' principles, from fundamental concepts to complex laws, further discussing real and ideal gases, solutions, electrochemical cells, and statistical thermodynamics. It

concludes with an examination of the kinetic theory of gases, transport phenomena, and chemical kinetics.

- FUNDAMENTALS OF PHYSICS, HALLIDAY ET AL. [37] (fund) covers undergraduate physics topics, ranging from fundamental concepts like motion and energy to more advanced areas such as quantum physics and nuclear physics.

- ELEMENTARY DIFFERENTIAL EQUATIONS AND BOUNDARY VALUE PROBLEMS, [45] (diff) provides a detailed exploration of differential equations, progressing from basic mathematical models to advanced topics like the Laplace Transform, linear systems, numerical methods, and Fourier series. It culminates with a deep dive into nonlinear equations, partial differential equations, and boundary value problems.

- PROBABILITY AND STATISTICAL INFERENCE, [44] (stat) covers probability and statistics, including fundamental concepts, discrete and continuous distributions, bivariate distributions, functions of random variables, and estimation techniques.

- CALCULUS: EARLY TRANSCENDENTALS, [43] (calculus) begins with diagnostic tests in foundational topics, and explores functions from multiple perspectives. It comprehensively covers calculus concepts from limits to three-dimensional analytic geometry, incorporating applications in various fields.

## A.2 EXAMINATION

- INTRODUCTION TO DATA MINING provides an introductory survey of data mining, which involves the automatic discovery of patterns, associations, changes, and anomalies in large databases. It explores various application areas of data mining, including bioinformatics, e-commerce, environmental studies, financial markets, multimedia data processing, network monitoring, and social service analysis.

- FUNDAMENTALS ARTIFICIAL INTELLIGENCE provides an introduction to the core problem-solving and knowledge representation paradigms in artificial intelligence. It covers Lisp programming with regular assignments, as well as topics such as search methods, planning techniques, knowledge structures, natural language processing, expert systems, vision, and parallel architectures.

- DIFFERENTIAL EQUATIONS covers various topics in differential equations, including first-order and second-order linear equations with constant coefficients, power series solutions, and linear systems. Students will explore the principles and applications of these mathematical concepts.

## A.3 TEXTBOOK EXAMPLES

# B SCIBENCH: MORE STATISTICS

## B.1 UI DESIGN

We employed a team of seven individuals to gather data from textbooks using an annotation tool. Each individual was responsible for 1-2 books, encompassing approximately 100 examples. The user interface of the annotation tool is depicted in Figure S2. For subsequent verification, we preserved images of problems and their corresponding answers. To ensure clarity in future references, we have maintained the original sequence of problems as they appear in the textbooks.

# C EXPERIMENTAL DETAILS

## C.1 PROMPTING

ChatGPT and GPT-4's API have three message parameters: SYSTEM, USER, and ASSISTANT. The SYSTEM parameter represents the system prompt, which provides context and instructions to the model. The USER parameter is the training prompt or input provided by the user, and the ASSISTANT parameter contains the model's output or response. We provide all system prompts and training prompts used in our experiments as below.

**Problem** (`fund`)
Two charged particles are fixed to an $x$ axis: Particle 1 of charge $q_1 = 2.1 \times 10^{-8}$C is at position $x = 20$ cm and particle 2 of charge $q_2 = -4.00q_1$ is at position $x = 70$ cm. At what coordinate on the axis (other than at infinity) is the net electric field produced by the two particles equal to zero?
**Answer:** $-30$ cm

**Problem** (`thermo`)
$N_2O_3(g)$ dissociates according to the equilibrium $N_2O_3(g) \rightleftharpoons NO_2(g) + NO(g)$. At 298 K and one bar pressure, the degree of dissociation defined as the ratio of moles of $NO_2(g)$ or $NO(g)$ to the moles of the reactant assuming no dissociation occurs is $3.5 \times 10^{-3}$. Calculate $\Delta G_R^\circ$ for this reaction.
**Answer:** 28 kJ mol$^{-1}$

**Problem** (`class`)
Halley's comet, which passed around the sun early in 1986, moves in a highly elliptical orbit with an eccentricity of 0.967 and a period of 76 years. Calculate its minimum distances from the Sun.
**Answer:** $8.8 \times 10^{10}$ m

**Problem** (`quan`)
A one-particle, one-dimensional system has $\Psi = a^{-1/2}e^{-|x|/a}$ at $t = 0$, where $a = 1.0000$ nm. At $t = 0$, the particle's position is measured. Find the probability that the measured value is between $x = 0$ and $x = 2$ nm.
**Answer:** 0.4908

**Problem** (`chemmc`)
One of the most powerful modern techniques for studying structure is neutron diffraction. This technique involves generating a collimated beam of neutrons at a particular temperature from a high-energy neutron source and is accomplished at several accelerator facilities around the world. If the speed of a neutron is given by $v_n = (3k_BT/m)^{1/2}$, where $m$ is the mass of a neutron, then what temperature is needed so that the neutrons have a de Broglie wavelength of 50pm ?
**Answer:** 2500 K

**Problem** (`atkins`)
The change in molar internal energy when $CaCO_3(s)$ as calcite converts to another form, aragonite, is $+0.21$ kJ mol$^{-1}$. Calculate the difference between the molar enthalpy and internal energy changes when the pressure is 1.0 bar given that the densities of the polymorphs are 2.71 g cm$^{-3}$ and 2.93 g cm$^{-3}$, respectively.
**Answer:** -0.28 Pa m$^3$ mol$^{-1}$

**Problem** (`matter`)
In an industrial process, nitrogen is heated to 500 K at a constant volume of 1.000 m$^3$. The gas enters the container at 300 K and 100 atm. The mass of the gas is 92.4 kg. Use the van der Waals equation to determine the approximate pressure of the gas at its working temperature of 500 K. For nitrogen, $a = 1.39$dm$^6$ atm mol$^{-2}$, $b = 0.0391$dm$^3$ mol$^{-1}$.
**Answer:** 140 atm

**Problem** (`calc`)
A planning engineer for a new alum plant must present some estimates to his company regarding the capacity of a silo designed to contain bauxite ore until it is processed into alum. The ore resembles pink talcum powder and is poured from a conveyor at the top of the silo. The silo is a cylinder 100ft high with a radius of 200ft. The conveyor carries ore at a rate of $60,000\pi$ ft$^3$/h and the ore maintains a conical shape whose radius is 1.5 times its height. If, at a certain time $t$, the pile is 60ft high, how long will it take for the pile to reach the top of the silo?
**Answer:** 9.8 h

**Problem** (`stat`)
In a study concerning a new treatment of a certain disease, two groups of 25 participants in each were followed for five years. Those in one group took the old treatment and those in the other took the new treatment. The theoretical dropout rate for an individual was 50% in both groups over that 5 -year period. Let $X$ be the number that dropped out in the first group and $Y$ the number in the second group. Assuming independence where needed, give the sum that equals the probability that $Y \geq X + 2$. HINT: What is the distribution of $Y - X + 25$ ?
**Answer:** 0.3359

**Problem** (`diff`)
Newton's law of cooling states that the temperature of an object changes at a rate proportional to the difference between its temperature and that of its surroundings. Suppose that the temperature of a cup of coffee obeys Newton's law of cooling. If the coffee has a temperature of $200°$F when freshly poured, and 1 min later has cooled to $190°$F in a room at $70°$F, determine when the coffee reaches a temperature of $150°$F
**Answer:** 6.07 min

Figure S1: Textbook examples with acronym highlighted in brown.

**System Prompt for Zero-Shot, Few-Shot, and Chain-of-Thought setting**:
Please provide a clear and step-by-step solution for a scientific problem in the categories of Chemistry, Physics, or Mathematics. The problem will specify the unit of measurement, which should not be included in the answer. Express the final answer as a decimal number with three digits after the decimal point. Conclude the answer by stating "The answer is therefore \boxed[ANSWER]."

**System Prompt for Python setting**:
Please provide a clear and step-by-step solution for a scientific problem in the categories of Chemistry, Physics, or Mathematics. The problem will specify the unit of measurement. Please translate the solution steps into Python code and encase the Python code within triple backticks for clarity.

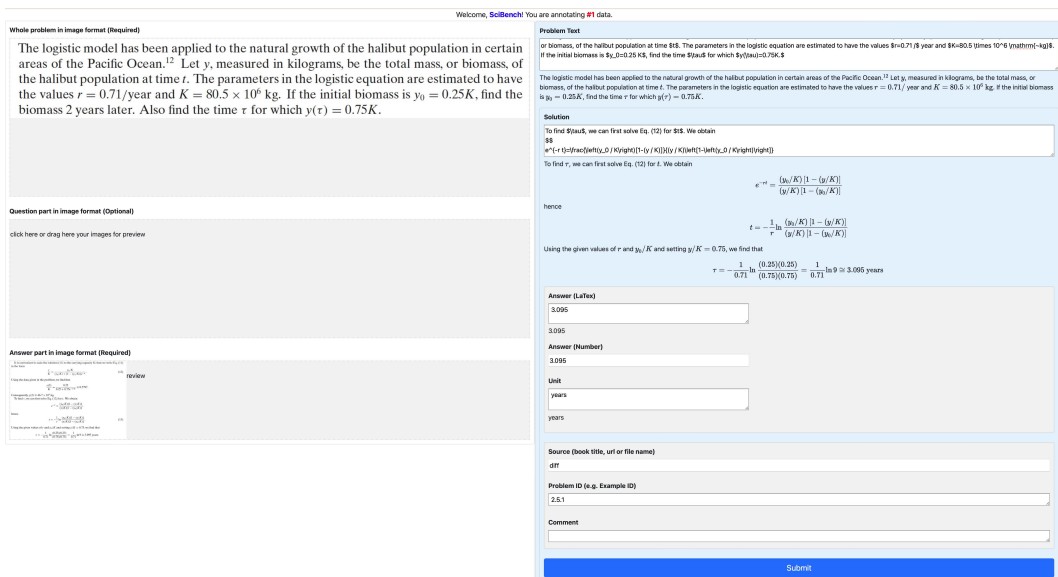

Figure S2: The UI design of data annotation.

**System Prompt for Wolfram setting**:
Please provide a clear and step-by-step solution for a scientific problem in the categories of Chemistry, Physics, or Mathematics. The problem will specify the unit of measurement. Please translate the solution steps into Wolfram code and encase the Wolfram Language code within triple backticks for clarity.

**System Prompt for Evaluation Protocol**:
Examine the given problem, the correct solution, and the model's solution. Identify the reason for the error in the model's solution based on the following 10 categories:
1. Logical Decomposition and Analysis Skills: This ability involves decomposing the problem into smaller, manageable parts, and understanding the relationships between these parts.
2. Identification of Assumptions: This skill involves the AI's ability to recognize relevant and necessary assumptions in the problem.
3. Spatial Perception: This is important for understanding problems in areas such as physics and chemistry, where you need to visualize molecules, forces, fields, etc.
4. Causal Reasoning: This is the ability to understand cause and effect relationships.
5. Problem Deduction Skills: This pertains to the ability to infer and deduce potential solutions or underlying principles from the given information in a problem.
6. Abstract Reasoning: This skill involves the ability to understand complex concepts that can't be perceived physically, and to recognize patterns or relationships beyond concrete examples.
7. Scientific Literacy: This skill involves a comprehensive understanding of key scientific principles, terminology, and methodologies across a range of disciplines.
8. Code Conversion Skills: This denotes the ability to accurately translate solution steps into different programming languages, like Python or Wolfram, without syntax errors.
9. Logical Reasoning: This is the ability to make a reasoned argument and to identify fallacies or inconsistencies in an argument or set of data.
10. Calculation Skills: This involves the ability to accurately carry out mathematical operations and computations.
Conclude your final error reason category number within \boxed.

**Training Prompt for Zero-Shot Chain-of-Thought**:
*Stage 1:*
Input: [input-question] Let's think step by step.
Output: <explanation>
*Stage 2:*
Input: [input-question] Let's think step by step. [explanation] + Therefore, the answer is:

Output: <answer>

**Training Prompt for Few-Shot**:
Input:
Problem 1: [Question 1] The answer is \boxed{[Answer 1]}.
Problem 2: [Question 2] The answer is \boxed{[Answer 2]}.
...
Problem n: [Question n] The answer is \boxed{[Answer n]}.
Problem n+1: [Question n+1]
Output: The answer is \boxed{<answer>}.

**Training Prompt for Few-Shot Chain-of-Thought**:
Input:
Problem 1: [Question 1] Explanation for Problem 1: [Explanation 1]. The answer is \boxed{[Answer 1]}.
Problem 2: [Question 2] Explanation for Problem 2: [Explanation 2]. The answer is \boxed{[Answer 2]}.
...
Problem n: [Question n] Explanation for Problem n: [Explanation n]. The answer is \boxed{[Answer n]}.
Problem n+1: [Question n+1]
Output: Explanaiton for Problem n+1: <explanation>. The answer is \boxed{<answer>}.

**Training Prompt for Few-Shot Python/Wolfram**:
Input:
Problem 1: [Question 1] Explanation for Problem 1: [Explanation 1]. Python/Wolfram language for Problem 1: ```[Python/Wolfram code 1]```.
Problem 2: [Question 2] Explanation for Problem 2: [Explanation 2]. Python/Wolfram language for Problem 2: ```[Python/Wolfram code 2]```.
...
Problem n: [Question n] Explanation for Problem n: [Explanation n]. Python/Wolfram language for Problem n: ```[Python/Wolfram code n]```.
Problem n+1: [Question n+1]
Output: Explanaiton for Problem n+1: <explanation>. Python/Wolfram language for Problem n+1: ```[Python/Wolfram code n+1]```.

**Training Prompt for Evaluation Protocol**:
Input: The question is [input-question]. The correct solution is [Correct-Solution]. The model solution is [Model-Solution].
Output: <Error Type>

**Training Prompt for Evaluation Protocol in Python/Wolfram**:
Input: The question is [input-question]. The correct solution is [Correct-Solution]. The model solution is [Model-Solution]. The translated program generates the answer as [Program Generated Answer], which is treated as model's output answer.
Output: <Error Type>

## C.2 EXPERIMENT PROCESS

All model output is extracted using \boxed{} notation. To prevent any missed extractions, we supplement this process with a manual check. For both Python and Wolfram settings, we extract the programming language with the triple backtick ```method, subsequently executing it within the corresponding language. The entirety of our code can be accessed via the following URL: https://anonymous.4open.science/r/anonymous-4FFB.

## D PROBLEM SOLVING ABILITIES OF CURRENT LLMS

### D.1 EXAMPLE

**Problem**
Calculate the de Broglie wavelength of an electron traveling at $1.00\%$ of the speed of light.

**Correct Solution**
The mass of an electron is $9.109 \times 10^{-31}$ kg. One percent of the speed of light is

$$v = (0.0100) \left( 2.998 \times 10^8 \text{ m} \cdot \text{s}^{-1} \right) = 2.998 \times 10^6 \text{ m} \cdot \text{s}^{-1}$$

The momentum of the electron is given by

$$p = m_{\text{e}} v = \left( 9.109 \times 10^{-31} \text{ kg} \right) \left( 2.998 \times 10^6 \text{ m} \cdot \text{s}^{-1} \right) = 2.73 \times 10^{-24} \text{ kg} \cdot \text{m} \cdot \text{s}^{-1}$$

The de Broglie wavelength of this electron is

$$\lambda = \frac{h}{p} = \frac{6.626 \times 10^{-34} \text{ J} \cdot \text{s}}{2.73 \times 10^{-24} \text{ kg} \cdot \text{m} \cdot \text{s}^{-1}} = 2.43 \times 10^{-10} \text{ m} = 243 \text{pm}$$

This wavelength is of atomic dimensions.
**Final Answer:** 243

**Chatgpt Solution with Chain-of-Thought Method**
Let's start by recalling the de Broglie wavelength formula: $\lambda = \frac{h}{p}$ where $\lambda$ is the de Broglie wavelength, $h$ is Planck's constant, and $p$ is the momentum of the particle. To calculate the momentum of the electron, we can use the relativistic momentum formula:

$$p = \frac{m_e v}{\sqrt{1 - \frac{v^2}{c^2}}}$$

where $m_e$ is the mass of the electron, $v$ is its velocity, and $c$ is the speed of light. Plugging in the given values, we get:

$$p = \frac{(9.11 \times 10^{-31} \text{kg})(0.01c)}{\sqrt{1 - (0.01c)^2/c^2}} = 8.19 \times 10^{-23} \text{kg} \cdot \text{m/s}$$

Now we can plug this momentum into the de Broglie wavelength formula:

$$\lambda = \frac{h}{p} = \frac{6.626 \times 10^{-34} \text{J} \cdot \text{s}}{8.19 \times 10^{-23} \text{kg} \cdot \text{m/s}} = 8.09 \text{pm}$$

Therefore, the de Broglie wavelength of the electron is $\boxed{8.09}$.
**Final Answer:** 8.09

**ChatGPT Solution without Chain-of-Thought Method**
The de Broglie wavelength of a particle is given by the equation: $\lambda = \frac{h}{p}$ where $h$ is the Planck constant and $p$ is the momentum of the particle. The momentum of an electron is given by:

$$p = mv$$

where $m$ is the mass of the electron and $v$ is its velocity. We are given that the electron is traveling at $1.00\%$ of the speed of light, which is: $v = 0.01c$ where $c$ is the speed of light. The mass of an electron is: $m = 9.11 \times 10^{-31}$, kg Putting it all together, we have:

$$\lambda = \frac{h}{mv} = \frac{h}{m(0.01c)}$$

Substituting the values of $h$, $m$, and $c$, we get:

$$\lambda = \frac{6.626 \times 10^{-34}, \text{J} \cdot \text{s}}{9.11 \times 10^{-31}, \text{kg} \cdot 0.01 \cdot 2.998 \times 10^8, \text{m/s}} = 3.31 \times 10^{-12} \text{m} = 3.31 \text{pm}$$

Therefore, the de Broglie wavelength of the electron is $\boxed{3.31 \text{pm}}$
**Final Answer:** 3.31

Figure S3: An example problem, inaccurately solved by ChatGPT with and without the Chain of Thought method, is provided to highlight different types of errors. In the example, the mistaken step is highlighted in red.

In the context of each specific capability, we present several exemplary errors accompanied by their corresponding classifications and explanations derived from the GPT model. Referencing Figure S3, the ChatGPT solution employing the Chain-of-Thought method corresponds to error category 4, "Causal Reasoning". The model explanation posits that "The error reason category is 4. Causal Reasoning. The model solution uses the relativistic momentum formula to calculate the momentum of the electron, which is not necessary for this problem since the electron is traveling at only $1.00\%$

of the speed of light. The relativistic momentum formula is only needed when the velocity of the particle is a significant fraction of the speed of light. Therefore, using this formula in this problem leads to an incorrect value for the momentum and consequently an incorrect value for the de Broglie wavelength." Meanwhile, the ChatGPT solution that doesn't utilize the Chain-of-Thought method is categorized under error case 10, "Calculation Skills". This example is further highlighted as an illustrative showcase for explanations of the Chain-of-Thought method in Section 5. More examples resulting in different error types are shown in Figure S4, S5, S6, S7, S8, S9, S10, S11, S12,S13.

---

**Problem**
Find the length of the cardioid $r = 1 + \sin\theta$.

---

**Correct Solution**
The cardioid's full length is given by the parameter interval $0 \leq \theta \leq 2\pi$, so Formula 5 gives

$$L = \int_0^{2\pi} \sqrt{r^2 + \left(\frac{dr}{d\theta}\right)^2}\, d\theta = \int_0^{2\pi} \sqrt{(1+\sin\theta)^2 + \cos^2\theta}\, d\theta = \int_0^{2\pi} \sqrt{2 + 2\sin\theta}\, d\theta$$

We could evaluate this integral by multiplying and dividing the integrand by $\sqrt{2 - 2\sin\theta}$, or we could use a computer algebra system. In any event, we find that the length of the cardioid is $L = 8$.
**Final Answer:** 8

---

**Model Solution**
To find the length of the cardioid, we use the formula for arc length in polar coordinates:

$$L = \int_\alpha^\beta \sqrt{r^2 + \left(\frac{dr}{d\theta}\right)^2}\, , d\theta,$$

where $\alpha$ and $\beta$ are the angles that correspond to one full loop of the cardioid (i.e. $\alpha = 0$ and $\beta = 2\pi$). First, we find $\frac{dr}{d\theta}$:

$$\frac{dr}{d\theta} = \frac{d}{d\theta}(1 + \sin\theta) = \cos\theta.$$

Then, we plug in $r = 1 + \sin\theta$ and $\frac{dr}{d\theta} = \cos\theta$ into the formula for arc length:

$$L = \int_0^{2\pi} \sqrt{(1+\sin\theta)^2 + \cos^2\theta}$$
$$d\theta = \int_0^{2\pi} \sqrt{1 + 2\sin\theta + \sin^2\theta + \cos^2\theta}$$
$$d\theta = \int_0^{2\pi} \sqrt{2 + 2\sin\theta}$$
$$d\theta = 4\int_0^{\pi} \sqrt{2 + 2\sin\theta}$$
$$d\theta \quad \text{(due to symmetry)} = 8\int_0^{\pi/2} \sqrt{2 + 2\sin\theta}$$
$$d\theta = 8\int_0^{\pi/2} \sqrt{4\cos^2\frac{\theta}{2}}$$
$$d\theta \quad \text{(using the half-angle formula)} = 16\int_0^{\pi/2} \cos\frac{\theta}{2}$$
$$d\theta = 32\sin\frac{\theta}{2}\Big|_0^{\pi/2}$$
$$= 32.$$

Therefore, the length of the cardioid is 32, and the unit of the answer is whatever unit was used for $r$ (e.g. meters, centimeters, etc.).
**Final Answer:** 32

---

**Error Reason**
The model solution made an error in the calculation of the integral $\int_0^{2\pi} \sqrt{2 + 2\sin\theta}\, d\theta$. The correct solution correctly simplified this integral to $4\int_0^{\pi} \sqrt{2 + 2\sin\theta}\, d\theta$ due to the symmetry of the integrand, and then used a substitution to evaluate the integral. The model solution, on the other hand, incorrectly evaluated this integral as $8\pi$. This error falls under the category of $\boxed{\text{10. Calculation Skills}}$.

Figure S4: An example problem is inaccurately solved by error reason 10. Calculation Skills. The mistaken step is highlighted in red.

## D.2 ASSESSMENT OF EVALUATION PROTOCOL

In order to assess the effectiveness of our evaluation protocol's classification, we enlisted the assistance of two annotators to determine whether the errors identified by the model verifier were accurate or

---

**Problem**
Use Simpson's Rule with $n = 10$ to approximate $\int_1^2 (1/x) dx$.

---

**Correct Solution**
Putting $f(x) = 1/x$, $n = 10$, and $\Delta x = 0.1$ in Simpson's Rule, we obtain

$$\int_1^2 \frac{1}{x} dx$$
$$\approx S_{10}$$
$$= \frac{\Delta x}{3} [f(1) + 4f(1.1) + 2f(1.2) + 4f(1.3) + \cdots + 2f(1.8) + 4f(1.9) + f(2)]$$
$$= \frac{0.1}{3} \left( \frac{1}{1} + \frac{4}{1.1} + \frac{2}{1.2} + \frac{4}{1.3} + \frac{2}{1.4} + \frac{4}{1.5} + \frac{2}{1.6} + \frac{4}{1.7} + \frac{2}{1.8} + \frac{4}{1.9} + \frac{1}{2} \right)$$
$$\approx 0.693150$$

**Final Answer:** 0.693150

---

**Model Solution**
Explanation for Problem 6: With $n = 10$ we have $h = \frac{2-1}{10} = \frac{1}{10}$ and $x_i = 1 + ih$ for $i = 0, 1, 2, \ldots, 10$. Then Simpson's Rule gives

$$\int_1^2 \frac{1}{x} dx \approx \frac{h}{3} [f(x_0) + 4f(x_1) + 2f(x_2) + 4f(x_3) + \ldots + 2f(x_8) + 4f(x_9) + f(x_{10})]$$

where $f(x) = \frac{1}{x}$.
wolfram language for Problem 6:
(* Constants *)
n = 10;
a = 1;
b = 2;
(* Function to integrate *)
f[x_] := 1/x;
(* Simpson's Rule *)
h = (b - a)/n;
x = Table[a + i*h, i, 0, n];
approximation = (h/3)*Total[Table[If[EvenQ[i], 2, 4]*f[x[[i]]], i, 1, n - 1] + f[x[[0]]], f[x[[n]]], 4*f[x[[n/2]]]];
approximation
**Final Answer:** "Objects of unequal length in {4,20/11,10/3,20/13,20/7,4/3,5/2,20/17,20/9}+{1/List,10/19,20/7} cannot be combined."

---

**Error Reason**
The error reason category for the model's solution is $\boxed{\text{8. Code Conversion Skills}}$ . The model's solution contains a syntax error in the Wolfram Language code, which causes the program to exceed the recursion limit and terminate prematurely. This error likely occurred during the translation of the correct solution's code into Wolfram Language.

Figure S5: An example problem is inaccurately solved by error reason 8. Code Conversion Skills. "Error Reason" denotes the output from the LLM Verifier utilized in the classification of error causes. In the example, the mistaken step is highlighted in red.

---

**Problem**
Use Simpson's Rule with $n = 10$ to approximate $\int_1^2 (1/x) dx$.

---

**Correct Solution**
The equations for $c_1$ and $c_2$ are

$$c_1(\alpha - E) + c_2\beta = 0 \quad \text{and} \quad c_1\beta + c_2(\alpha - E) = 0$$

For $E = \alpha + \beta$, either equation yields $c_1 = c_2$. Thus, $\psi_b = c_1(2p_{z1} + 2p_{z2})$. The value of $c_1$ can be found by requiring that the wave function be normalized. The normalization condition on $\psi_\pi$ gives $c_1^2(1 + 2S + 1) = 1$. Using the Hückel assumption that $S = 0$, we find that $c_1 = 1/\sqrt{2}$. Substituting $E = \alpha - \beta$ into either of the equations for $c_1$ and $c_2$ yields $c_1 = -c_2$, or

$$\psi_a = c_1(2p_{z1} - 2p_{z2})$$

The normalization condition gives $c^2(1 - 2S + 1) = 1$, or $c_1 = 1/\sqrt{2}$.
**Final Answer:** 0.7071067

---

**Model Solution**
therefore not provided as this problem requires visual representations and cannot be solved through text alone.

---

**Error Reason**
The error reason category is $\boxed{\text{3. Spatial Perception}}$ . This problem requires visual representations to find the bonding and antibonding Hückel molecular orbitals for ethene. The model's solution is incomplete as it only provides equations and does not provide any visual representation.

Figure S6: An example problem is inaccurately solved by error reason 3. Spatial Perception. "Error Reason" denotes the output from the LLM Verifier utilized in the classification of error causes.

not. Through the annotation of 151 samples across different settings, we observed that 123 of them were correctly classified, resulting in an accuracy rate of 81.45%.

---

**Problem**
What is the mean speed, $\bar{c}$, of $N_2$ molecules in air at $25\,^\circ C$

---

**Correct Solution**
The integral required is

$$\bar{c} = 4\pi \left(\frac{M}{2\pi RT}\right)^{3/2} \int_0^\infty v^3 e^{-Mv^2/2RT}\, dv \quad = 4\pi \left(\frac{M}{2\pi RT}\right)^{3/2} \times \frac{1}{2}\left(\frac{2RT}{M}\right)^2 = \left(\frac{8RT}{\pi M}\right)^{1/2}$$

where we have used the standard result from tables of integrals (or software) that

$$\int_0^\infty x^3 e^{-ax^2}\, dx = \frac{1}{2a^2}$$

Substitution of the data then gives

$$\bar{c} = \left(\frac{8 \times \left(8.3141\ \mathrm{J\ K^{-1}\ mol^{-1}}\right) \times (298\ \mathrm{K})}{\pi \times \left(28.02 \times 10^{-3}\ \mathrm{kg\ mol^{-1}}\right)}\right)^{1/2} = 475\ \mathrm{m\ s^{-1}}$$

where we have used $1\ \mathrm{J} = 1\ \mathrm{kg\ m^2\ s^{-2}}$.
**Final Answer:** 475

---

**Model Solution**
approximately $515\ \mathrm{m\ s^{-1}}$. To calculate this, we can use the formula for the root-mean-square (rms) speed of gas molecules:

$$c_{\mathrm{rms}} = \sqrt{\frac{3kT}{m}}$$

where $k$ is the Boltzmann constant, $T$ is the temperature in Kelvin, and $m$ is the mass of the molecule. For $N_2$, $m = 28$ amu. Plugging in the values, we get:

$$c_{\mathrm{rms}} = \sqrt{\frac{3 \times 1.38 \times 10^{-23}\ \mathrm{J/K} \times (25 + 273)\ \mathrm{K}}{28 \times 1.66 \times 10^{-27}\ \mathrm{kg/amu}}} \approx 515\ \mathrm{m\ s^{-1}}$$

Note that we converted the temperature from Celsius to Kelvin and the mass from atomic mass units (amu) to kilograms.
**Final Answer:** 515

---

**Error Reason**
The error reason category is $\boxed{\text{7. Scientific Literacy}}$. The model solution uses the formula for the root-mean-square (rms) speed of gas molecules, which is only valid for ideal gases. However, the given problem does not specify that the gas is ideal, and in fact, air is not an ideal gas. The correct solution uses a more general formula that applies to all gases, including non-ideal gases like air. Therefore, the model's solution is not applicable to the given problem.

---

Figure S7: An example problem is inaccurately solved by error reason 7. Scientific Literacy. "Error Reason" denotes the output from the LLM Verifier utilized in the classification of error causes. In the example, the mistaken step is highlighted in red, while the correct step is highlighted in blue.

---

**Problem**
What is the number of possible four-letter code words, selecting from the 26 letters in the alphabet?

---

**Correct Solution**
The number of possible four-letter code words, selecting from the 26 letters in the alphabet, in which all four letters are different is

$$_{26}P_4 = (26)(25)(24)(23) = \frac{26!}{22!} = 358,800.$$

**Final Answer:** 358800

---

**Model Solution**
Explanation for Problem 6: The number of possible four-letter code words, selecting from the 26 letters in the alphabet is

$$26^4 = 456,976.$$

**Final Answer:** 456976

---

**Error Reason**
The model's error reason category is $\boxed{\text{9. Logical Reasoning}}$. The model did not consider the condition that all four letters should be different. The correct solution considers this condition and uses the permutation formula to calculate the number of possible four-letter code words.

---

Figure S8: An example problem is inaccurately solved by error reason 9. Logical reasoning. "Error Reason" denotes the output from the LLM Verifier utilized in the classification of error causes. In the example, the mistaken step is highlighted in red.

## D.3 COMPARISON

| **Problem** |
| --- |
| Find the bonding and antibonding Hückel molecular orbitals for ethene. |

| **Correct Solution** |
| --- |
| The equations for $c_1$ and $c_2$ associated with Equation 11.7 are

$$c_1(\alpha - E) + c_2\beta = 0 \quad \text{and} \quad c_1\beta + c_2(\alpha - E) = 0$$

For $E = \alpha + \beta$, either equation yields $c_1 = c_2$. Thus,

$$\psi_b = c_1\left(2p_{z1} + 2p_{z2}\right)$$

The value of $c_1$ can be found by requiring that the wave function be normalized. The normalization condition on $\psi_\pi$ gives $c_1^2(1 + 2S + 1) = 1$. Using the Hückel assumption that $S = 0$, we find that $c_1 = 1/\sqrt{2}$. Substituting $E = \alpha - \beta$ into either of the equations for $c_1$ and $c_2$ yields $c_1 = -c_2$, or

$$\psi_a = c_1\left(2p_{z1} - 2p_{z2}\right)$$

The normalization condition gives $c^2(1 - 2S + 1) = 1$, or $c_1 = 1/\sqrt{2}$.
**Final Answer:** 0.70710678 |

| **Model Solution** |
| --- |
| two bonding and two antibonding Hückel molecular orbitals for ethene. The unit of the answer is not applicable as it is a qualitative description of the molecular orbitals.
**Final Answer:** None |

| **Error Reason** |
| --- |
| The error reason category for the model's solution is $\boxed{\text{6. Abstract Reasoning}}$. The model's solution is incorrect because it did not consider the Hückel molecular orbital theory, which predicts that there are only two pi molecular orbitals for ethene: one bonding and one antibonding. This is due to the fact that ethene has only two pi electrons, which can occupy the two pi molecular orbitals. Therefore, the model's solution of two bonding and two antibonding molecular orbitals is not possible for ethene. |

Figure S9: An example problem is inaccurately solved by error reason 6. Abstract Reasoning. "Error Reason" denotes the output from the LLM Verifier utilized in the classification of error causes.

| **Problem** |
| --- |
| A living room has floor dimensions of $3.5$ m and $4.2$ m and a height of $2.4$ m. What does the air in the room weigh when the air pressure is $1.0$ atm ? |

| **Correct Solution** |
| --- |
| The air's weight is equal to $mg$, where $m$ is its mass Mass $m$ is related to the air density $\rho$ and the air volume $V$ ($\rho = m/V$). Calculation: Putting the two ideas together and taking the density of air at 1.0 atm, we find

$$mg = (\rho V)g \quad = \left(1.21 \text{ kg/m}^3\right)(3.5 \text{ m} \times 4.2 \text{ m} \times 2.4 \text{ m})\left(9.8 \text{ m/s}^2\right) = 418 \text{ N}$$

**Final Answer:** $418$ |

| **Model Solution** |
| --- |
| $101, 640$ N. To find the weight of the air in the room, we need to first calculate the volume of the room: $V = lwh = 3.5 \text{ m} \times 4.2 \text{ m} \times 2.4 \text{ m} = 35.28 \text{ m}^3$ Next, we can use the ideal gas law to find the mass of the air in the room: $PV = nRT$ where $P$ is the pressure, $V$ is the volume, $n$ is the number of moles of gas, $R$ is the gas constant, and $T$ is the temperature. Assuming room temperature of $20^\circ$C (293 K), we can solve for $n$:

$$n = \frac{PV}{RT} = \frac{(1.0 \text{ atm})(35.28 \text{ m}^3)}{(0.0821 \text{ L atm/mol K})(293 \text{ K})} = 1,610 \text{ mol}$$

Finally, we can find the weight of the air using the molar mass of air (approximately 28.97 g/mol):

$$W = mg = nMg = (1,610 \text{ mol})(0.02897 \text{ kg/mol})(9.81 \text{ m/s}^2) = 101,640 \text{ N}$$

**Final Answer:** $101640$ |

| **Error Reason** |
| --- |
| The error in the model's solution is that they used the ideal gas law to find the mass of the air in the room, which assumes that the air behaves like an ideal gas. However, the ideal gas law is not applicable in this case because the problem does not provide information about the temperature of the air. The correct approach is to use the density of air at 1.0 atm and the volume of the room to find the mass of the air, as shown in the correct solution. Therefore, the reason for the error in the model's solution is $\boxed{\text{2. Identification of Assumptions}}$. |

Figure S10: An example problem is inaccurately solved by error reason 2. Identification of Assumptions. "Error Reason" denotes the output from the LLM Verifier utilized in the classification of error causes. In the example, the mistaken step is highlighted in red.

**Problem**
From an ordinary deck of playing cards, cards are to be drawn successively at random and without replacement. What is the probability that the third spade appears on the sixth draw?

**Correct Solution**
Let $A$ be the event of two spades in the first five cards drawn, and let $B$ be the event of a spade on the sixth draw. Thus, the probability that we wish to compute is $P(A \cap B)$. It is reasonable to take

$$P(A) = \frac{\left( \begin{array}{c} 132 \end{array} \right)\left( \begin{array}{c} 39 \\ 3 \end{array} \right)}{\left( \begin{array}{c} 525 \end{array} \right)} = 0.274 \quad \text{and} \quad P(B \mid A) = \frac{11}{47} = 0.234$$

The desired probability, $P(A \cap B)$, is the product of those numbers: $P(A \cap B) = (0.274)(0.234) = 0.064$
**Final Answer:** 0.064

**Model Solution**
The probability that the third spade appears on the sixth draw is

$$\frac{{}_{13}C_2 \cdot {}_{39}C_4}{{}_{52}C_6} = \frac{78 \cdot 741,948}{20,358,520} = \frac{1,441}{5,577}.$$

**Final Answer:** 0.258

**Error Reason**
The model's solution is incorrect because it calculates the probability of getting two spades in the first five cards and any spade on the sixth draw, instead of specifically calculating the probability of the third spade appearing on the sixth draw given that two spades have already appeared in the first five draws. Therefore, the error reason category is $\boxed{\text{4. Causal Reasoning}}$

Figure S11: An example problem is inaccurately solved by error reason 4. Causal Reasoning. The mistaken step is highlighted in red.

**Problem**
Gas cylinders of $CO_2$ are sold in terms of weight of $CO_2$. A cylinder contains 50lb (22.7 kg ) of $CO_2$. How long can this cylinder be used in an experiment that requires flowing $CO_2$ at 293 K($\eta = 146\mu P$) through a 1.00-m-long tube (diameter $= 0.75$ mm) with an input pressure of $1.05$ atm and output pressure of $1.00$ atm ? The flow is measured at the tube output

**Correct Solution**
The gas flow rate $\Delta V/\Delta t$ is
$$\frac{\Delta V}{\Delta t} = \frac{\pi r^4}{16\eta L P_0}\left(P_2^2 - P_1^2\right) = \quad 2.76 \times 10^{-6} \text{ m}^3 \text{ s}^{-1}$$

Converting the $CO_2$ contained in the cylinder to the volume occupied at 298 K and 1 atm pressure, we get

$$n_{CO_2} = 22.7 \text{ kg}\left(\frac{1}{0.044 \text{ kg mol}^{-1}}\right) = 516 \text{ mol}, V = \frac{nRT}{P} = 1.24 \times 10^4 \text{ L}\left(\frac{10^{-3} \text{ m}^3}{\text{L}}\right) = 12.4 \text{ m}^3$$

Given the effective volume of $CO_2$ contained in the cylinder, the duration over which the cylinder can be used is $\frac{12.4 \text{ m}^3}{2.76 \times 10^{-6} \text{ m}^3 \text{ s}^{-1}} = 4.49 \times 10^6$ s. This time corresponds to roughly 52 days.
**Final Answer:** 52 days

**Model Solution**
The volume of $CO_2$ in the cylinder is given by

$$V = \frac{m}{\rho} = \frac{22.7 \text{ kg}}{1.98 \text{ kg m}^{-3}} = 11.5 \text{ m}^3$$

The volume of $CO_2$ that flows through the tube is given by

$$V = \frac{\pi}{4}d^2 L = \frac{\pi}{4}\left(0.75 \times 10^{-3} \text{ m}\right)^2 (1.00 \text{ m}) = 1.77 \times 10^{-7} \text{ m}^3$$

The time required to flow $V$ volume of $CO_2$ is given by $t = \frac{V}{Q} = \frac{VP_2}{Q_m P_1} = \frac{\left(1.77 \times 10^{-7} \text{ m}^3\right)(1.00 \text{ atm})}{(1.05 \text{ atm} - 1.00 \text{ atm})(146\mu P)} = 2.08 \times 10^6$ s. The number of seconds in a year is $3.15 \times 10^7$ s. Therefore, the cylinder can be used for $\frac{2.08 \times 10^6 \text{ s}}{3.15 \times 10^7 \text{ s yr}^{-1}} = 0.066 \text{ yr} = 24.1 \text{ days}$
**Final Answer:** 24.1 days

**Error Reason**
The model's solution is incorrect because it calculates the time required to flow a volume of $CO_2$ through the tube, but it does not take into account the total volume of $CO_2$ in the cylinder. The correct solution calculates the total volume of $CO_2$ in the cylinder and divides it by the flow rate to determine the duration over which the cylinder can be used. Therefore, the error reason category is $\boxed{\text{5. Problem Deduction Skills}}$.

Figure S12: An example problem is inaccurately solved by error reason 5. Problem Deduction Skills. The mistaken step is highlighted in red.

**Problem**

An astronaut whose height $h$ is 1.70 m floats "feet down" in an orbiting space shuttle at distance $r = 6.77 \times 10^6$ m away from the center of Earth. What is the difference between the gravitational acceleration at her feet and at her head?

**Correct Solution**

We can approximate Earth as a uniform sphere of mass $M_E$. Then, from Eq. 13-11, the gravitational acceleration at any distance $r$ from the center of Earth is

$$a_g = \frac{GM_E}{r^2}$$

We might simply apply this equation twice, first with $r = 6.77 \times 10^6$ m for the location of the feet and then with $r = 6.77 \times 10^6$ m + 1.70 m for the location of the head. However, a calculator may give us the same value for $a_g$ twice, and thus a difference of zero, because $h$ is so much smaller than $r$. Here's a more promising approach: Because we have a differential change $dr$ in $r$ between the astronaut's feet and head, we should differentiate Eq. 13-15 with respect to $r$. Calculations: The differentiation gives us

$$da_g = -2\frac{GM_E}{r^3}dr$$

where $da_g$ is the differential change in the gravitational acceleration due to the differential change $dr$ in $r$. For the astronaut, $dr = h$ and $r = 6.77 \times 10^6$ m. Substituting data into Eq.13-16, we find

$$da_g = -2\frac{\left(6.67 \times 10^{-11} \text{ m}^3/\text{kg} \cdot \text{s}^2\right)\left(5.98 \times 10^{24} \text{ kg}\right)}{(6.77 \times 10^6 \text{ m})^3}(1.70 \text{ m}) \quad = -4.37 \times 10^{-6} \text{ m/s}^2$$

**Final Answer:** $-4.37 \times 10^{-6}$ m/s$^2$

**Model Solution**

We can use the equation for the gravitational acceleration due to a point mass, given by Eq. 13-2 ($g = GM/r^2$), to find the gravitational acceleration at the astronaut's feet and head. Because the astronaut is in free fall, the difference between these two accelerations is the tidal acceleration, given by Eq. 13-22 ($\Delta g = 2GMh/r^3$). Calculations: The gravitational acceleration at the astronaut's feet is

$$g_{\text{feet}} = \frac{GM}{r^2} = \frac{\left(6.67 \times 10^{-11} \text{ N} \cdot \text{m}^2/\text{kg}^2\right)\left(5.98 \times 10^{24} \text{ kg}\right)}{(6.77 \times 10^6 \text{ m})^2} = 9.78 \text{ m/s}^2.$$

The gravitational acceleration at the astronaut's head is

$$g_{\text{head}} = \frac{\color{red}{GM}}{\color{red}{(r + h)^2}} = \frac{\left(6.67 \times 10^{-11} \text{ N} \cdot \text{m}^2/\text{kg}^2\right)\left(5.98 \times 10^{24} \text{ kg}\right)}{(6.77 \times 10^6 \text{ m} + 1.70 \text{ m})^2} = 9.77 \text{ m/s}^2.$$

The difference between these two accelerations is

$$\Delta g = g_{\text{feet}} - g_{\text{head}} \quad = 9.78 \text{ m/s}^2 - 9.77 \text{ m/s}^2 = 0.01 \text{ m/s}^2 \quad = 1.0 \times 10^{-5} \text{ m/s}^2.$$

**Final Answer:** $1.0 \times 10^{-5}$ m/s$^2$

**Error Reason**

The model's solution is incorrect because it uses the equation for the gravitational acceleration due to a point mass, which is not applicable in this case since the Earth is not a point mass. The correct equation to use is the one given in the correct solution, which takes into account the fact that the Earth is a uniform sphere of mass. Therefore, the error reason category is | 1. Logical Decomposition and Analysis Skills |.

Figure S13: An example problem is inaccurately solved by error reason 1. Logical Decomposition and Analysis Skills. "Error Reason" denotes the output from the LLM Verifier utilized in the classification of error causes. In the example, the mistaken step is highlighted in red.

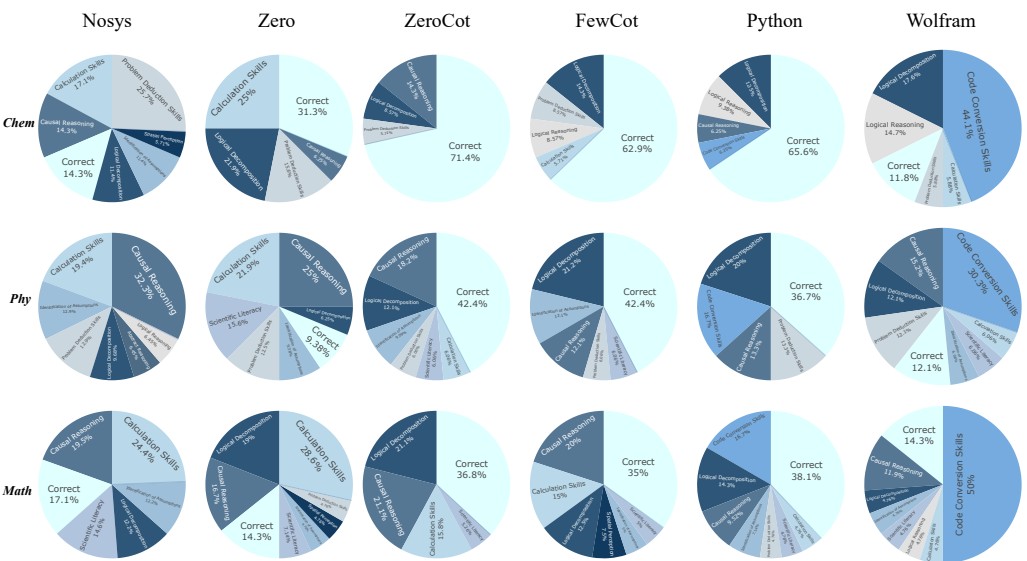

Figure S14: Comparison of error reasons between different subjects. The qualitative analysis of incorrect answers provided by the model revealed ten dimensions of problem-solving abilities. The "Correct" category, representing the proportion of accurately answered questions by the model, is included for comparison. A smaller chart representation indicates a lower number of erroneously answered questions.

---

**Problem**
$3x + 4y = -23, 2y - x = -19$. What is the solution $(x, y)$ to the system of equations above?
Options: (A). (-5,2) (B). (3,-8) (C). (4,-6) (D). (9,-6)
**Final Answer:** B

---

**Problem**
What is the mean speed, $\bar{c}$, of $N_2$ molecules in air at $25°C$
**Correct Solution**
The integral required is

$$\bar{c} = 4\pi \left( \frac{M}{2\pi RT} \right)^{3/2} \int_0^\infty v^3 e^{-Mv^2/2RT} \, dv \quad = 4\pi \left( \frac{M}{2\pi RT} \right)^{3/2} \times \frac{1}{2} \left( \frac{2RT}{M} \right)^2 = \left( \frac{8RT}{\pi M} \right)^{1/2}$$

where we have used the standard result from tables of integrals (or software) that

$$\int_0^\infty x^3 e^{-ax^2} \, dx = \frac{1}{2a^2}$$

Substitution of the data then gives

$$\bar{c} = \left( \frac{8 \times \left( 8.3141 \, \text{J K}^{-1} \, \text{mol}^{-1} \right) \times (298 \, \text{K})}{\pi \times \left( 28.02 \times 10^{-3} \, \text{kg mol}^{-1} \right)} \right)^{1/2} = 475 \, \text{m s}^{-1}$$

where we have used $1 \, \text{J} = 1 \, \text{kg m}^2 \, \text{s}^{-2}$.
**Final Answer:** 475

---

Figure S15: Example from AGIEval [18] (top) and SCIBENCH (bottom). The problem from AGIEval is of high school difficulty with basic algebraic computations involved. In contrast, the problem from our dataset is of a college-level complexity, requiring not only an understanding of background equations but also proficiency in differentiation calculations.

