# OpenReview forum: "SciBench: Evaluating College-Level Scientific Problem-Solving Abilities of Large Language Models"
_ICLR.cc/2024/Conference — Submitted to ICLR 2024_

### Official Review · Reviewer_Satc · 2023-10-30

**Soundness:** 3 good
**Presentation:** 3 good
**Contribution:** 3 good
**Rating:** 6
**Confidence:** 4

**Summary:**

This paper provides a benchmark, SciBench, for solving scientific problems. The previous benchmark datasets for measuring the ability of large language models (LLMs) were for the middle school or high school subjects, and only contain multiple-choice questions. The authors collect two datasets, one open-set and one closed-set, containing college-level scientific questions. Through this dataset, the authors also provide an in-depth discussion of the current inability of LLMs and how they mistake.

**Strengths:**

- This benchmark is more challenging than the previous related benchmarks, and annotations and varied evaluation methods are provided.
- The evaluation result with multiple LLMs is exhaustive and suggestive to the community.
- The paper also provides LLM-based error analysis for incorrect responses, which is an interesting way as a novel benchmarking approach.

**Weaknesses:**

- The experiments report a combination of external tools and few-shot learning. On the other hand, Table 4 shows that the accuracy of few-shot learning is sometimes lower than that of zero-shot learning. The reviewer would like to know why the authors did not try the combination of external tools and zero-shot learning.
- In the experiment, if the LLM responses were within a relative error of 0.05 as a numerical value, they were considered to be correct. On the other hand, as shown in the center of Figure 1, there are cases where the numerical calculation is wrong even though the idea or formula itself is correct, and cases where the numerical calculation is correct by chance but the idea or formula is false. In this evaluation setup, the former case is simply considered wrong, while the latter case is correct. It is questionable whether this is a suitable way to evaluate scientific problem-solving ability.

**Questions:**

- The reviewer expects the authors to respond to the points listed in Weaknesses.
- Section 3 says "All problems are carefully verified by human annotators to ensure that LaTeX documents can be compiled without any syntax errors." The reviewer wonders if the authors do not check for parsing errors other than syntax errors.
- In section 5, it seems that LLM is only applied to numerically incorrect examples for error analysis. What would be the result if the same error analysis was applied to answers that are numerically correct? This involves two aspects:
  - First, it is possible to find numerically correct answers that contain some errors listed in section 5.
  - Also, it would be even better if there was also an evaluation of error analysis for numerically correct responses, while it has been reported that about 20% of error analysis by LLM for numerically incorrect responses are discarded.

---

> ### Author Response · Authors · 2023-11-21
>
> Thank you for your positive reviews and insightful comments. Please find our detailed response below.
>
> **W1**: The experiments report a combination of external tools and few-shot learning. On the other hand, Table 4 shows that the accuracy of few-shot learning is sometimes lower than that of zero-shot learning. The reviewer would like to know why the authors did not try the combination of external tools and zero-shot learning.
>
> **R1**: Thank you for bringing this question up. We had concerns about the ability of LLM to adapt programming languages, especially Wolfram Language, in the zero-shot setting. However, we conducted the experiment on the zero-shot python setting using gpt-3.5-turbo and show the result below.
> | atkins | cal   | chemmc | class | diff  | fund  | matter | quan  | stat  | thermo | avg |
> |--------|-------|--------|-------|-------|-------|--------|-------|-------|--------|---------|
>  | 11.21  | 19.04 | 30.77  | 0.0   | 14.00 | 9.59  | 2.04   | 11.76 | 42.67 | 4.48   | 14.75   |
>
> The result is lower than the 19.91% achieved in the few-shot setting. We will further conduct experiments with a combination of zero-shot setting and external tools.
>
> **W2**: In the experiment, if the LLM responses were within a relative error of 0.05 as a numerical value, they were considered to be correct. On the other hand, as shown in the center of Figure 1, there are cases where the numerical calculation is wrong even though the idea or formula itself is correct, and cases where the numerical calculation is correct by chance but the idea or formula is false. In this evaluation setup, the former case is simply considered wrong, while the latter case is correct. It is questionable whether this is a suitable way to evaluate scientific problem-solving ability.
>
> **R2**: Problems with wrong numerical calculation are considered as wrong since calculation ability is considered one of the problem-solving ability. Models should have the calculation ability to answer problems correctly.  Considering the overall difficulty of our tested problems, it has a very low probability that the model derives the answer correctly but with wrong formulas.
> Unlike multiple-choice questions, where random guessing has a high chance of yielding the correct answer, our evaluation method is more stringent as it involves a wide range of rational numbers, making it highly improbable to guess the correct answer randomly. Our confidence in the evaluation metric is based on the belief that if the model arrives at the correct answer, it is very likely due to a correct solution process, given the complexity involved in reaching these answers. Our answers, which can be as specific as -0.0987 or as large as 1036592783, are difficult to guess accurately.
> Implementing a process verifier could enhance the quality of evaluation. However, employing human annotators for this task could significantly increase project costs. Our evaluation criteria are consistent with most existing benchmarks, like GSM8K[1], where the assessment is based solely on the numerical equivalence of the final answer. We also include further error analysis for inaccurately answered questions in Section 5.
>
> **Q2**: Section 3 says "All problems are carefully verified by human annotators to ensure that LaTeX documents can be compiled without any syntax errors." The reviewer wonders if the authors do not check for parsing errors other than syntax errors.
>
> **A2**: Thanks for your questions. Every type of error, including parse errors, is thoroughly checked, and we employ human annotators for final verification in the last phase of the process.
>
> **Q3**: In section 5, it seems that LLM is only applied to numerically incorrect examples for error analysis. What would be the result if the same error analysis was applied to answers that are numerically correct? This involves two aspects:
>
> First, it is possible to find numerically correct answers that contain some errors listed in section 5.
> Also, it would be even better if there was also an evaluation of error analysis for numerically correct responses, while it has been reported that about 20% of error analysis by LLM for numerically incorrect responses are discarded.
>
> **A3**: Thanks for your question. We have manually checked the correct answer of LLM and didn’t detect any wrongly classified examples.
>
> [1] Cobbe, K., Kosaraju, V., Bavarian, M., Chen, M., Jun, H., Kaiser, L., ... & Schulman, J. (2021). Training verifiers to solve math word problems. arXiv preprint arXiv:2110.14168.

---

> > ### Comment · Reviewer_Satc · 2023-11-23
> >
> > As a result of the authors' responses, the reviewer retains the initial score.
> >
> > R2 is not very convincing; GSM8K is only an elementary school math problem and uses numerical equivalence as an evaluation of it. The title of this paper mentions that the proposed dataset is a benchmark for scientific problem-solving at college-level. It is unlikely that the difference between arithmetic for elementary school students and scientific problem-solving for college students is simply a difference in the number of reasoning steps, and it would be more in line with the subjective view if the evaluation could be based on something other than final value. The authors could add a discussion of R1 and R2 to the main paper or the appendix for future work.

---

> > > ### Author Response · Authors · 2023-11-23
> > >
> > > Thanks for your response. We want to make some clarifications:
> > >
> > > 1. We are aware that the difference between arithmetic for elementary school students and scientific problem-solving for college students is **not in the number of reasoning steps**. In addition, it involves background knowledge and more **advanced calculations**, including integration and differentiation. Our reason for the current evaluation metric is that if the model can provide the correct final answer, it is highly likely that it has utilized accurate background knowledge and performed the calculations correctly. Compared to other advanced dataset benchmarks like AGIEval[1], which uses multiple-choice format, our evaluation metric more effectively reveals the accuracy of the problem-solving ability of a model by using the free-response format. We have shown an illustrative comparison between AGIEval and SciBench in Appendix Figure S15, where using multiple-choice format might allow LLM to guess the correct answer from the options but free-response format will much lower the possibility.
> > >
> > > 2. Incorporating other evaluating metrics might be helpful. In Section 5, we showed an analysis using LLM to analyze the solution process made by the original LLM and identifying the corresponding lacking abilities, which further analyzes the problem solving abilities of LLM.
> > >
> > > 3. We will further have a comprehensive discussion about R1 and R2.
> > >
> > > [1] Zhong, W., Cui, R., Guo, Y., Liang, Y., Lu, S., Wang, Y., ... & Duan, N. (2023). Agieval: A human-centric benchmark for evaluating foundation models. arXiv preprint arXiv:2304.06364.

---

### Official Review · Reviewer_Rk3o · 2023-11-01

**Soundness:** 3 good
**Presentation:** 3 good
**Contribution:** 3 good
**Rating:** 8
**Confidence:** 4

**Summary:**

This paper introduces an interesting and valuable benchmark dataset called SCIBENCH. The dataset contains challenging college-level problems in physics, chemistry, and mathematics, as well as detailed solutions, which facilitates detailed analysis of model performance. The inclusion of free-response questions rather than just multiple-choice further increases the difficulty and better tests true reasoning skills.

**Strengths:**

The dataset is comprehensive. And the experiments compare several representative LLMs on SCIBENCH under different prompting strategies like chain-of-thought, zero-shot learning, and using Python/Wolfram tools. The results demonstrate SCIBENCH's ability to differentiate model capacities, with the top score of only 35.8% by GPT-4, highlighting room for improvement.

**Weaknesses:**

The paper is well-written, and the methodology is sound.  The only problem is that the name of SCIBENCH may be little overclaiming, with problems focused only on physics, chemistry, and math.

**Questions:**

The paper claims SCIBENCH problems require "multiple steps of reasoning." Is it possible to quantify the complexity and difficulty of the dataset? For example, the GSM8k dataset has 3~4 steps in solutions, which is a mathematical reasoning dataset.

The essential skill set was defined with just two human annotators. Was any inter-annotator agreement analysis performed to ensure consistency?

---

> ### Author Response · Authors · 2023-11-21
>
> Thank you very much for your positive comments and constructive reviews! We have the following response for your questions.
>
> **W1**: The paper is well-written, and the methodology is sound. The only problem is that the name of SCIBENCH may be little overclaiming, with problems focused only on physics, chemistry, and math.
>
> **R1**: Thank you for your valuable comments. Currently SciBench covers the most important domains in natural sciences. We plan to further expand our dataset involving more scientific domains.
>
>
> **Q1**: The paper claims SCIBENCH problems require "multiple steps of reasoning." Is it possible to quantify the complexity and difficulty of the dataset? For example, the GSM8k dataset has 3~4 steps in solutions, which is a mathematical reasoning dataset.
>
> **A1**: Thank you for your suggestions. SciBench is a college-level problem set from various subjects. Therefore, the notion of difficulty might be very subjective. Given the wide scope of subjects covered, it is also challenging to assemble a consistent cohort of annotators familiar with all topics to label the difficulty levels. However, to provide a sense of the problem complexity, we show the number of sentences for problems with step-by-step solutions.
> | Sentences   | % of Problems |
> |-------------|---------------|
> | n < 5       | 18.75%        |
> | 5 <= n < 10  | 26.79%        |
> | 10 <= n < 20 | 45.53%        |
> | n => 20      | 8.93%         |
>
> **Q2**: The essential skill set was defined with just two human annotators. Was any inter-annotator agreement analysis performed to ensure consistency?
>
> **A2**: Thank you for your question. Two annotators, by scanning through all error answers made by LLM, independently identify  the essential skills, and then they collaborate to merge and analyze these skills to arrive at the final results.

---

### Official Review · Reviewer_aRBB · 2023-11-07

**Soundness:** 2 fair
**Presentation:** 3 good
**Contribution:** 2 fair
**Rating:** 5
**Confidence:** 4

**Summary:**

This paper presents SciBench, a benchmark that includes two new curated datasets, one of problems extracted from college-level textbooks in subjects like mathematics, chemistry, and physics, and another of problems from undergraduate exams in computer science and mathematics. The work evaluates five LLMs with different prompting methods and categorizes different problem-solving abilities and errors made by LLMs.

**Strengths:**

1. The high-level categorization into different types of abilities and errors contributes to benchmarking in the field.

2. The authors highlight the limitations of previous models and methods.

3. The paper is well-written.

**Weaknesses:**

1. The work evaluates five LLMs; however is critically missing GPT-4 with code interpreter (ADA). The results are now superseded by GPT-4 turbo.

2. The work lacks a comparison with state-of-the-art methods such as hypothesis search and refinement and the use of a code interpreter.

3. Textbooks used for the new dataset are available online in pdf format which is easily converted into text. It is unclear that the dataset does not consist of questions on which the models have already been trained on.

4. The selection criteria of the college textbooks and exams for the new datasets lack detailed explanation and justification.

5. The abilities and errors may be extended and refined by dividing into sub-categories.

**Questions:**

How many of the questions in the new datasets are available online by search? or in the latest LLMs?

---

> ### Author Response · Authors · 2023-11-21
>
> **W1**: The work evaluates five LLMs; however is critically missing GPT-4 with code interpreter (ADA). The results are now superseded by GPT-4 turbo.
>
> **R1**: Thanks for your information. Currently the API of code interpreters for GPT models is not accessible. Since as the reviewer mentioned, OPENAI has spent much effort on adding code support for their model, we conduct a new experiment with their newly released API, GPT4-Turbo, and results are shown below:
> |           | atkins | cal  | chemmc | class | diff | fund | matter | quan | stat | thermo | avg |
> |-----------|--------|------|--------|-------|------|------|--------|------|------|--------|---------|
> | gpt4-turbo| 28.0   | 23.8 | 20.5   | 21.3  | 36.0 | 31.5 | 16.3   | 35.3 | 53.3 | 9.0    | 28.3    |
> | gpt4      | 21.1   | 42.9 | 41.0   | 17.0  | 34.0 | 38.4 | 28.6   | 38.2 | 69.3 | 29.9   | 35.8    |
>
> As shown in the table, the newly released GPT4-Turbo performs slightly lower than the original GPT4 version.
>
>
> **W2**: The work lacks a comparison with state-of-the-art methods such as hypothesis search and refinement and the use of a code interpreter.
>
> **R2**: Thanks for your information. For the use of code interpreters, our experiment results under ‘CoT+Py’ and ‘CoT+Wol’  settings are the results that integrate LLM with external programmatic tools. Regarding some extensive search methods, many of them cannot be easily extended to general scientific problem-solving tasks. Among these methods we choose the more advanced self-consistency[1] method using gpt-3.5-turbo and show the results as below:
>  | atkins | chemmc | quan | matter | fund | class | thermo | diff | stat | calculus | avg   |
> |--------|--------|------|--------|------|-------|--------|------|------|----------|-------|
>  | 15.6   | 23.8   | 11.4 | 11.8   | 15.8 | 2.0   | 4.5    | 8.0  | 31.2 | 26.0     | 15.6  |
>
> We are exploring the other techniques as well and will further update our manuscript with these findings.
>
>
> **W3**: Textbooks used for the new dataset are available online in pdf format which is easily converted into text. It is unclear that the dataset does not consist of questions on which the models have already been trained on.
>
> **R3**: Most of the problems and their corresponding answers in textbooks are located in separate sections. For example, the problems might be found on Page 100, while the solutions are located much later, for instance, on Page 695. We employed several human annotators to manually extract questions and answers and aligned them together, which is not trivial work. This layout makes it challenging for Large Language Models (LLMs) to correlate them without specialized extraction methods.
>
>
> **W4**: The selection criteria of the college textbooks and exams for the new datasets lack detailed explanation and justification.
>
> **R4**: We choose the most representative and widely used textbooks as our dataset. Most of them are assigned course material in college courses. Furthermore, those problem sets are suitable for evaluation purposes since they contain many practice problems and their answers are in numerical value.
>
>
> **W5**: The abilities and errors may be extended and refined by dividing into sub-categories.
>
> **R5**: Thank you for your information. The current abilities have general coverage for chemistry, physics and math domains. It is hard to further differentiate them. However, we show the lacking ability results for different sub-categories in Appendix D. By checking the solution steps made by LLM, code conversion ability can be further divided into sub-reasons like syntax errors, translation errors. We will do the analysis in the future to further investigate this area.
>
>
> **Q1**: How many of the questions in the new datasets are available online by search? or in the latest LLMs?
>
> **A1**: We conducted a random search in our dataset and found that none of the problems together with solutions can be directly searched from search engines .
>
> [1] Wang, X., Wei, J., Schuurmans, D., Le, Q., Chi, E., Narang, S., ... & Zhou, D. (2022). Self-consistency improves chain of thought reasoning in language models. arXiv preprint arXiv:2203.11171.

---

### Official Review · Reviewer_evdV · 2023-11-08

**Soundness:** 4 excellent
**Presentation:** 2 fair
**Contribution:** 2 fair
**Rating:** 3
**Confidence:** 5

**Summary:**

A new dataset consisting of 695 prompts is introduced to test how well several LLMs (GPT-4, Claude, LLaMA) perform in college-level scientific reasoning tasks: math, physics, chemistry. The authors evaluate along different assessment axes how well various approaches performed (e.g. Zero-Shot with CoT, Few-Shot with CoT etc.). They conclude that the mastery of problem solving ability remains weak.

**Strengths:**

- a good selection of LLMs was used
- state of the art methods, such as CoT is used
- LLMs are allowed to use tools

**Weaknesses:**

- the name of the paper, SciBench, is similar to another dataset, called *BIG-bench* (https://arxiv.org/abs/2206.046159), which also overlaps partly with SciBench. Nonetheless BIG-bench is not cited.
- not quite novel impact of the dataset: Some important reasoning datasets on college-level (and beyond) were omitted from the literature review, e.g., *Mathematical Capabilities of ChatGPT* (https://arxiv.org/abs/2301.13867) and *NaturalProofs* (https://arxiv.org/abs/2104.01112) for math, which both already do some of the things SciBench proposed to achieve (see page 4, "Enabling of assessing advanced problem solving ability" which both mentioned papers achieve; and "Inclusion of college-level problems" and "Inaccessibility in text formats" which the first reference paper achieves.)
The *Mathematical Capabilities of ChatGPT* paper also essentially collects error profiles, as the authors do in Figure 3, as does the *NaturalProver* paper (https://arxiv.org/pdf/2205.12910.pdf), not to be confused with *NaturalProofs*
- There should be a more detailed, in-text comparison with these datasets mentioned above to highlight similarities and differences of SciBench with these other, pertinent datasets. I also have some doubts (see *Questions* section), if Table 1 is really 100% accurate.
- 695 examples is a rather small dataset
- Section 5: "From 112 such error annotations and with the assistance of GPT-4, we distill these errors into ten essential skills that GPT-3.5 might lack". Using a LLM to make a decision is a source of errors. It would be best if no LLM were used - and if it is used, a very detailed explanation should be given of how exactly it provided assistance.

**Questions:**

- I don't understand how the annotation process works. Do the authors mean by that an evaluation of the output of the model (by humans)? Or do they mean that the existing input data into the LLM was augmented ("annotated")?
If the former was meant, then why does the MATH dataset in Table 1, in the Analysis column, under "Auto" have a "No"? This is incorrect, the MATH uses automatic evaluation by virtue of constraining the output in the \boxed{...} environment.

---

> ### Author Response · Authors · 2023-11-21
>
> Thank you for your careful reviews and detailed comments. Please find our responses to your concerns below.
>
> **W1**: the name of the paper, SciBench, is similar to another dataset, called BIG-bench (https://arxiv.org/abs/2206.046159), which also overlaps partly with SciBench. Nonetheless BIG-bench is not cited.
>
> **R1**: Thank you for your information. We added BIG-bench as a reference in Section 2. BIG-bench is different from SciBench because it mainly targets specific tasks like predicting element names or deducing  how physical mechanisms work, with some of its tasks as high-school level with multiple-choice questions. On the other hand, SciBench uses harder college-level free-response problems.
>
> **W2**: not quite novel impact of the dataset: Some important reasoning datasets on college-level (and beyond) were omitted from the literature review, e.g., Mathematical Capabilities of ChatGPT (https://arxiv.org/abs/2301.13867) and NaturalProofs (https://arxiv.org/abs/2104.01112) for math, which both already do some of the things SciBench proposed to achieve (see page 4, "Enabling of assessing advanced problem solving ability" which both mentioned papers achieve; and "Inclusion of college-level problems" and "Inaccessibility in text formats" which the first reference paper achieves.)
>
> **R2**: Thank you for mentioning this question. The mentioned datasets focus solely on mathematics.  Our dataset includes problems from different disciplines such as chemistry, physics, and computer science, which require specific in-domain background knowledge for their solution process, rather than purely mathematical questions that rely solely on reasoning ability. As far as our knowledge, there are significantly fewer corpus and evaluation benchmarks about other scientific subjects compared with mathematics.
>
> **W3**: There should be a more detailed, in-text comparison with these datasets mentioned above to highlight similarities and differences of SciBench with these other, pertinent datasets. I also have some doubts (see Questions section), if Table 1 is really 100% accurate.
>
> **R3**:  Thank you for your question. The key difference of our approach from others lies in the use of a dataset comprising college-level questions that require complex reasoning processes and computational steps such as differentiation. We choose free-response answers over multiple-choice formats. Furthermore, we employ various configurations of Language Learning Models (LLM) like Chain of Thought, or external tools, to evaluate our dataset. Additionally, we utilize an automated error analysis pipeline for categorizing errors.   For Table1, the column “Level”, “computation”, “solution”, “type” denote the Dataset referring to Section 3. Column “zero-shot”, “few-shot”, “CoT”, “tool” denote the experiment referring to Section 4. Column “Human” and “auto” denote the analysis referring to Section 5. Evaluation part represents if other benchmark work has been tested using those configurations. Analysis represents if other benchmark works use manual or automatic analysis for their dataset.  We further include an in-text comparison in Appendix D, Figure S15.
>
> **W4**: 695 examples is a rather small dataset
>
> **R4**: Thanks for bringing up this question. Our dataset goes through human extraction for each problem, which is highly costly. Our goal for the dataset is to serve as an evaluation dataset for problem solving ability. We're currently using the same annotation pipeline to scale up the dataset.
>
>
> **W5**: Section 5: "From 112 such error annotations and with the assistance of GPT-4, we distill these errors into ten essential skills that GPT-3.5 might lack". Using a LLM to make a decision is a source of errors. It would be best if no LLM were used - and if it is used, a very detailed explanation should be given of how exactly it provided assistance.
>
> **R5**: Thank you for your question. Two human annotators participate in the process. Decisions on the final abilities are determined by annotators, aided by assistants. By going through errors, these two annotators develop ten abilities and then employ a Language Learning Model (LLM) as a third evaluator to suggest additional abilities. They then compare and refine their findings based on this input. Ultimately, the final outcomes are determined by the annotators. After LLM annotate the error reasons, we conduct human-check by sampling 151 examples across all settings to make sure the annotations make sense. We make this human-AI cooperated analysis pipeline to reduce the cost of human post-analysis, while incorporate human checking to make sure the correctness of LLM decision and try to reduce the risk that reviewer mentioned. Though not perfect, we believe it can be another type of analyzing framework for future study of LLM problem-solving.

---

> ### Author Response · Authors · 2023-11-21
>
> **Q1**: I don't understand how the annotation process works. Do the authors mean by that an evaluation of the output of the model (by humans)? Or do they mean that the existing input data into the LLM was augmented ("annotated")? If the former was meant, then why does the MATH dataset in Table 1, in the Analysis column, under "Auto" have a "No"? This is incorrect, the MATH uses automatic evaluation by virtue of constraining the output in the \boxed{...} environment.
>
> **A1**: In Table 1, there are sections titled "Evaluation" and "Analysis." In the "Evaluation" section, we evaluate the performance of LLM under different settings. Following that, in the "Analysis" section, we analyze the specific problem solving abilities where each setting falls short. Under the "Analysis" section, "Human" denotes the use of human annotators to analyze the results produced by Large Language Models (LLMs), while "Auto" signifies the use of an automated process for this analysis. We have developed an evaluation protocol that automatically categorizes errors made by LLMs into specific lacking abilities.

---

> ### Comment · Reviewer_evdV · 2023-11-23
> **Reviewer answer**
>
> Thank you for the detailed responses. Below I am adding my feedback to the revised paper.
>
> R1: I am happy that you have added a remark regarding BIG-bench dataset, since the names really are very similar.
>
> R2: "*the mentioned datasets focus solely on mathematics. Our dataset includes problems from different disciplines such as chemistry, physics, and computer science, which require specific in-domain background knowledge for their solution process, rather than purely mathematical questions that rely solely on reasoning ability.*"
> I can't agree with this claim for a number of reasons:
>
> 1. **at least a third of the data is math** Table 2 outlines the exact components of your dataset of 695 examples (I am noticing now that it seems that currently, the total size of the dataset is nowhere mentioned and has to be computed from Table 2, as I did; the size of the dataset is one of the most important information and should be mentioned in a central place), out of which 221 are physics, 272 are chemistry and 202 are math. Thus, math questions make up almost half of your dataset, so I think dismissing this by stating that your dataset actually focuses on other disciplines.
>
> 2. **physics and chemistry are heavily grounded in math**: problems in physics/chemistry at this level can be essentially split up into two components: a modeling component, where the physical problem statement is translated correctly into a mathematical problem by using the correct existing physical formulae, and a mathematical component where the equations are solved. The first problem from the `fund_sol.json` file from the anonymized repository illustrates this perfectly. Hence, mathematical reasoning also overlaps with the other datasets.
>
> 3. **no separate computer science questions**: I am not sure where "computer science" questions are contained in your dataset since Table 2 lists none of them.
>
> R3: "The key difference of our approach from others lies in the use of a dataset comprising college-level questions that require complex reasoning processes and computational steps such as differentiation. We choose free-response answers over multiple-choice formats." These properties also apply to the two illustrative datasets I mentioned, so I would not say that these are the distinctive properties of your approach. To illustrate, here are further papers along these lines (also not cited): *Sparks of AGI* (https://arxiv.org/abs/2303.12712), *Evaluating the Logical Reasoning Ability of ChatGPT and GPT-4* (https://arxiv.org/pdf/2304.03439.pdf) whose question types partly overlap with the ones from the current paper.
>
> R4: I realize that human annotation is costly, but I accept that at this time, not much can be done. It would have been good to let the public know whether the dataset will be further grown or not.
>
> R5: Thank you for this information. Given these facts, I am satisfied that the process is sound. I would recommend the authors include this information in the paper, too, since it will aid readers.
>
> All in all, I was hoping that a more substantial revision of the paper would have been made that addressed and alleviated my concerns above.
> In particular, this would have meant:
> - at least indicating a path to increase the dataset (R4 above). It doesn't even have to be a dramatic increase since I do realize that this is costly, but some examples that cover more physical and chemistry questions, where the modeling part instead of the mathematical is more predominant, would have been good since this seems to me to really be the distinguishing feature of this dataset.
> - highlighting what the contribution really is (R2 and R3) above. This paper *Who Answers It Better? An In-Depth Analysis of ChatGPT and Stack Overflow Answers to Sofware Engineering Qestions* (https://arxiv.org/pdf/2308.02312.pdf) also has an error analysis section.
>
> Thus, on the contrary, my concerns have actually increased. Furthermore, I went through the anonymized dataset and compared some of the questions with those from other papers. It seems to me that the current paper is more of a re-versioning of previous articles, than a novel contribution. Also, it seems the authors are not aware of a substantial piece of the previous literature on reasoning-related datasets. I would say the part of the dataset that deals with LLMs selecting the correct equations to solve the physical and chemistry questions is the novel part of the paper - *this should have been more emphasized*.
>
> It seems that only two sentences (in red) have been added in the current paper revision, which seems a bit meager. At least some of the information the authors provided here in their rebuttal is highly relevant to the paper (see R5 above). This, unfortunately, leads me to re-adjust my score. I would hope that a full revision of the paper is made, to clearly outline how this dataset actually differs from previous ones and to highlight the novelty of the modeling part.

---

### Official Review · Reviewer_K8pK · 2023-11-09

**Soundness:** 3 good
**Presentation:** 3 good
**Contribution:** 3 good
**Rating:** 6
**Confidence:** 4

**Summary:**

This paper introduces SCIBENCH, a benchmark suite aimed at addressing the challenges associated with the reasoning capabilities of language models when solving complex scientific problems. SCIBENCH consists of two datasets: an open set containing collegiate-level scientific problems from mathematics, chemistry, and physics textbooks, and a closed set comprising problems from undergraduate-level exams in computer science and mathematics. The authors conduct a benchmarking study using five representative language models, employing zero-shot, few-shot, COT, and tool-augmented prompting strategies. This bench is challenge and the current state-of-the-art model (GPT-4) exhibit unsatisfactory performance, achieving only a 35.8% overall score on the benchmark. Furthermore, the paper identifies ten problem-solving abilities and categorizes the errors made by language models accordingly. The analysis demonstrates that no single prompting strategy consistently outperforms others, and there is a trade-off between improving certain skills and hindering others.

**Strengths:**

1. Proposing a reliable dataset to evaluate the reasoning ability of existing language models is crucial. This dataset proposed by the authors has two main advantages:

* The dataset is highly challenging, and the performance of GPT-4 is comparatively poor. The questions are all free-form, which helps prevent the model from result guesses based on multiple-choice answers. Furthermore, the dataset can effectively differentiate between different large language models (LLMs). (Open-source models like LLaMA-2-70B exhibit significantly inferior performance when compared to their closed-source counterparts.)

* The construction of this dataset avoids the issue of data leakage during the testing process. Throughout the dataset's construction, measures were taken to ensure that the questions should not be readily accessible online and cannot be easily extracted or transformed into text.


2. In the analysis section, the author further defines ten skills involved in the reasoning process and provides a detailed analysis of error cases.

3. To assess the reasoning ability of language models, this dataset serves as an good supplementary resource and is a valuable resource for the community.

**Weaknesses:**

The analysis section could be further enhanced. For instance, regarding the observation "The zero-shot learning setting exhibits comparable performance to the few-shot learning setting," did the author conduct comprehensive testing on different in-context examples to ascertain consistent findings? Different qualities of few-shot examples may yield different results. High-quality prompts and in-context examples should help stimulate the model's capabilities.

**Questions:**

1. Why does utilizing prompts from Wolfram hinder the capabilities of the model? What specific attempts have been made, and could it be due to inadequate adjustments to the prompts? It would be more appropriate to refrain from drawing such a statement without further investigation.

2. On page8, third line from the bottom, is it written incorrectly? Is "15.2% of casual ability" referring to the zero-shot setting?

3. Few-shot prompting leads to a trade-off in skills. Is this due to the content of the example prompts? Can it be resolved by replacing few-shot examples?

---

> ### Author Response · Authors · 2023-11-21
>
> We thank you very much for your detailed reviews and questions. For your main concerns, we would like to make the following clarifications.
>
> **W1**: The analysis section could be further enhanced. For instance, regarding the observation "The zero-shot learning setting exhibits comparable performance to the few-shot learning setting," did the author conduct comprehensive testing on different in-context examples to ascertain consistent findings? Different qualities of few-shot examples may yield different results. High-quality prompts and in-context examples should help stimulate the model's capabilities.
>
> **R1**: Thanks to the reviewer for bringing the question. For the in-context examples, we randomly selected them from the problems with solutions. We provide all prompt and in-context examples in Github Repo. Additionally, we conducted four experimental runs under the few-shot CoT setting using gpt-3.5-turbo. Each experiment used different randomly selected prompt examples, achieving an average result of 13.17% across all experiments. This 13.17% result, being higher than 11.99%, reflects the continuous updates to OpenAI's API. However, this figure is still comparable to the 12.17% seen in the zero-shot CoT setting, which supports our claim.
>
>
> **Q1**: Why does utilizing prompts from Wolfram hinder the capabilities of the model? What specific attempts have been made, and could it be due to inadequate adjustments to the prompts? It would be more appropriate to refrain from drawing such a statement without further investigation.
>
> **A1**: Thank you for highlighting this crucial question. As far as we know, Wolfram language is not appearing frequently in pre-training corpus compared to Python. Secondly, the grammar of Wolfram Language is complex and hard for LLM to learn from a few examples. Those two could be the reasons that using Wolfram Language hinders performance. For our project, we provided a detailed description of the Wolfram Language prompt in the Appendix C and Github Repo (https://anonymous.4open.science/r/anonymous-4FFB/dataset/original/wolfram). This prompt was crafted to align with the Python setup. We maintained consistency in the wording for both external tools, simply substituting "Python" with "Wolfram". While there may be potential for further accuracy enhancements, this approach was chosen to maintain fairness in comparing the two tools. To further improve the performance, we need to fine-tune LLM to learn how to use integrated tools, like ToRA[1], and we leave this for future work.
>
>
> **Q2**:On page8, third line from the bottom, is it written incorrectly? Is "15.2% of casual ability" referring to the zero-shot setting?
>
> **A2**: Thank you for bringing this up. This sentence means that the error rates of zero-shot setting are 15.2% for both causality ability and logical decomposition ability. We have rephrased this in the main paper for clarification.
>
>
> **Q3**: Few-shot prompting leads to a trade-off in skills. Is this due to the content of the example prompts? Can it be resolved by replacing few-shot examples?
>
> **A3**: Thanks to the reviewer for bringing the question. As shown in R1, we rigorously conduct multiple rounds of random selection of prompt examples and the result shows that LLM under the few-shot setting performs comparable to the zero-shot setting, which is consistent with our claim.  The reason behind this could be that the number of examples are not large enough to cover all possible abilities needed, and thus providing limited examples distracts the model.
>
> [1] Gou, Z., Shao, Z., Gong, Y., Yang, Y., Huang, M., Duan, N., & Chen, W. (2023). ToRA: A Tool-Integrated Reasoning Agent for Mathematical Problem Solving. arXiv preprint arXiv:2309.17452.

---

> > ### Comment · Reviewer_K8pK · 2023-11-23
> > **Thanks for your response**
> >
> > Thank you for the authors' response. It addressed several of my concerns.

---

### Author Response · Authors · 2023-11-21
**Thank you & summary of changes**

We sincerely thank all reviewers for their constructive comments and insightful suggestions. We are delighted that reviewers find that our dataset is reliable, valuable, challenging (Reviewer K8pK), and comprehensive(Reviewer Rk3o), and that our experimental study is exhaustive and suggestive(Reviewer Satc).

According to reviewers’ suggestions, we have revised our manuscript. Below, we outline the added experiments and highlight our revision in red color in the revised PDF.

- We conducted experiments on the few-shot setting with multiple runs of random selection and the combination of zero-shot setting and python.
- We included experiments using newly released API gpt4-turbo and advanced prompting strategy self-consistency.
- We added additional illustrative comparison examples between Scibench and other datasets.

Once again, we appreciate all reviewers for their valuable suggestions which enables us to further strengthen our work. We hope that our new revision resolves reviewers’ concerns and we are happy to answer further questions.

---

### Meta-Review · Area_Chair_9RHA · 2023-12-11

**Metareview:**

This work introduces a benchmark suite to evaluate the reasoning capabilities required for solving complex scientific problems. It receives mixed reviews. It is appreciated that the dataset contains challenging problems with detailed solutions, and the free-response questions further increases the difficulty. However, it covers only physics, chemistry, and math, and some statements deserve further investigation. Given there are already many related datasets/benchmarks on math and science, AC agrees that the novelty, scope and scale of the proposed benchmark should be significantly extended to fully justify its name and role as a systematic SCIBENCH.

**Justification For Why Not Higher Score:**

This benchmark covers only physics, chemistry, and math, and some statements deserve further investigation. Given there are already many related datasets/benchmarks on math and science, the novelty, scope and scale of the proposed benchmark should be significantly extended.

**Justification For Why Not Lower Score:**

N/A

---

### Decision · Program_Chairs · 2024-01-16

Reject